

# A Tethered And Navigated Air Blimp (TANAB) for observing the microclimate over a complex terrain

Manoj K. Nambiar[1], Ryan A. E. Byerlay[1], Amir Nazem[1], M. Rafsan Nahian[1], Mohsen Moradi[1], and Amir A. Aliabadi[1]

[1]School of Engineering, University of Guelph, Guelph, ON, Canada

**Correspondence:** Amir A. Aliabadi (aliabadi@uoguelph.ca)

**Abstract.** This study presents the first environmental monitoring field campaign of a newly developed Tethered And Navigated Air Blimp (TANAB) system to investigate the microclimate over a complex terrain. The use of a tethered balloon in complex terrains such as mines and tailings ponds is novel and the focus of the present study. The TANAB system was fully developed and launched at a mine facility in northern Canada in May 2018. This study describes the key design features, the sensor payload onboard, and the observations made by the TANAB system. The system measured meteorological conditions including wind speed in three directions, temperature, relative humidity, and pressure over the first few tens of meters of the atmospheric boundary layer. The system also performed earth surface thermal imaging, or temperature mapping, of the underlying surface. The measurements were made at two primary locations in the facility: i) near a tailings pond and ii) in a mine pit. TANAB measured the dynamics of the atmosphere at different diurnal times (e.g. day versus night) and locations (near tailings pond versus inside the mine). Such dynamics include mean and turbulence statistics pertaining to flow momentum and energy, and they are crucial in the understanding of emission fluxes from the facility in future studies. In addition, TANAB can provide boundary conditions and validation datasets to support mesoscale dispersion modelling or Computational Fluid Dynamics (CFD) simulations for various transport models.

## 1 Introduction

The Atmospheric Boundary Layer (ABL) is the lowest portion of the air near the earth surface that responds to surface processes in one hour or less (Stull, 1988; Aliabadi, 2018). The understanding of the atmospheric turbulent processes governing the transfer of heat, moisture, and momentum between the surface and the free troposphere is of practical importance for many applications such as weather and climate prediction, pollution dispersion modelling, and urban air quality studies (Zilitinkevich and Baklanov, 2002; Pichugina et al., 2008; Aliabadi et al., 2016b, c). Most of the research in atmospheric turbulence have been focused on relatively smooth terrain and horizontally homogeneous environments mainly due to the limitation in the availability of adequate observation platforms and difficulty in acquiring data from the complex environments. However, the study of the ABL and surface-atmosphere interaction over complex terrain is very important for many applications. Surface heterogeneity can cause horizontal gradients of momentum or temperature, and it can influence or complicate the horizontal and vertical transport mechanisms, for instance driven by slopes or thermals (Mahrt and Vickers, 2005; Medeiros and Fitzjarrald, 2014). In





addition, model parameterizations of turbulence processes established for atmospheric flows over smooth and homogeneous surfaces often fail when applied over inhomogeneous and complex terrains (Roth, 2000).

## 1.1 Technology Gaps

Two types of ABL observations of the meteorological parameters are key: atmospheric properties and earth surface properties (Mäkiranta et al., 2011; Manoj et al., 2014). Conventional techniques measuring the atmospheric properties such as remote sensing (e.g. satellite, RADARS[1], LIDARS[2], SODARS[3], radiometers) and in situ measurements (meteorological masts, aircraft, or sounding balloons) are widely-used for observing parameters such as wind, humidity, and temperature (Pichugina et al., 2008; Legain et al., 2013; Aliabadi et al., 2016b, 2018a). The main disadvantages of such conventional techniques are the low frequency of turbulence measurements (SODARs and LIDARs), cost (aircraft and satellites), difficulty of navigation (sounding balloons), intermittency of observation (Aircraft, sounding balloons, and non-geostationary satellites), low spatial resolution (geostationary satellites), and limited spatial coverage (meteorological towers) (Fernando and Weil, 2010; Medeiros and Fitzjarrald, 2014). In addition, measuring the surface layer within the ABL poses a serious challenge to aircraft that cannot fly at altitudes lower than 150 m in many jurisdictions for safety reasons.

As far as earth surface properties are concerned, Land Surface Temperature (LST) is a crucial meteorological parameter to be measured that influences the energy budget and dynamics of the atmosphere. It has historically been derived from conventional satellite-based sensors such as the MODerate resolution Imaging Spectroradiometer (MODIS) on the Terra and Aqua satellites, the Advanced Baseline Imager on the Geostationary Operational Environmental Satellites (GOES), the Enhanced Thematic Mapper Plus (ETM+) on Landsat 7 and the Thermal InfRared Sensor (TIRS) on Landsat 8 (Tomlinson et al., 2011; Chastain et al., 2019). These sensors however, only either record images with a high spatial resolution and a low temporal resolution (ETM+ and TIRS) or a high temporal resolution and a low spatial resolution (GOES) (Irons et al., 2012; Chastain et al., 2019; Cintineo et al., 2016; Schmit et al., 2017). Furthermore, satellite-based sensors are known to have missing and skewed data due to a variety of environmental factors including atmospheric effects, land surface emissivity and sensor failure (Li et al., 2013; Malamiri et al., 2018). Recent advancements of Unmanned Aerial Vehicles (UAVs) and miniaturization of thermal imaging technology, have allowed researchers to remotely sense the environment for more reliable and precise LST measurements (Malbéteau et al., 2018). With the inclusion of Inertial Measurement Units and Global Positioning Systems on UAVs, airborne images can be directly georeferenced without Ground Control Points (Turner et al., 2014). Quantitative data from images recorded from UAV systems can be readily derived from proprietary software including PhotoScan Professional, Pix4Dmapper and MATLAB (Verykokou and Ioannidis, 2018). Open source direct georeferencing and LST calculation software programs are not commonly used.

---

[1]Radio Detection And Ranging
[2]Light Detection And Ranging
[3]Sonic Detection And Ranging



## 1.2 Objectives

The Tethered And Navigated Air Blimp (TANAB), developed by the authors, is a unique mobile sensing platform for the investigation of surface layer within ABL overcoming some of the above limitations. TANAB is lifted by the buoyancy force and requires no propulsion power for navigation compared to drones. This allows turbulence measurements over long periods

without disturbing the surrounding air. It provides accurate in-situ measurements unlike remote sensing technologies such as LIDARs, SODARs, and satellites. It is safer to operate at low altitudes compared to manned aircraft. It can be navigated and redeployed using a tether, unlike radiosondes, and it is very cost effective. TANAB can collect high time-resolution observations of the weather to characterize the turbulence properties in low altitudes in almost all weather conditions. In case of extreme wind, TANAB can still be used with the help of additional stabilizing tethers (usually three). The variables it measures include

wind speed in three directions, temperature, relative humidity, and pressure. It also performs earth surface thermal imaging, or temperature mapping, of the underlying terrain.

TANAB observations can be utilized in numerous ways. Atmospheric dynamics as measured by TANAB can determine transport mechanisms that drive emission fluxes (Steudler et al., 1991). Factors such as mean wind speed, atmospheric diffusion coefficient, and thermal stability, greatly influence emission fluxes (Bowden et al., 1993), all of which are measured by

TANAB. It can also provide boundary conditions and validation datasets to support ABL simulations using Computational Fluid Dynamics (CFD) or mesoscale modelling. Furthermore, the environmental data collected can be used to develop and validate microclimate and dispersion models in complex urban environments (Krayenhoff and Voogt, 2007; Bueno et al., 2012; Holnicki and Nahorski, 2015; Aliabadi et al., 2017).

There are only a few comprehensive field studies that focus on the ABL over a mine environment while the structure of ABL

in an orographically complex terrain such as a mine can be complicated (Rotach and Zardi, 2007; Medeiros and Fitzjarrald, 2014, 2015). In this study, the TANAB system is demonstrated while being tested to reveal the surface layer meteorology of a complex terrain in a mining facility. In the surface layer, flows are highly influenced by the terrain geometry, while Coriolis effects have still negligible influences (Arroyo et al., 2014). TANAB measurements are reported at different diurnal times (e.g. day versus night) and locations (near tailings pond versus inside the mine).

## 1.3 Structure of the Paper

The structure of the paper is organized as follows. Section 2 briefly describes the TANAB system and the sensors payload. Calibration experiments are explained in Sect. 3. Section 4 presents field experiments and results of the environmental monitoring campaign where TANAB was used in a complex mining facility. Conclusions and future recommendations are provided in Sect. 5.





## 2 Tethered And Navigated Air Blimp (TANAB) Specification

TANAB consists of fixed and variable payloads. The fixed payload is comprised of a helium balloon, the navigation tether, a tether reel, and a gondola platform housing the sensors. The variable payload is comprised of microclimate sensors, such as a mini weather station and a thermal camera, and a flight controller. While the fixed payload is the same for every mission, the
5   variable payload can be altered to use different sensors suitable for a particular application.

### 2.1 Envelop and Platform

The balloon envelop is manufactured by Aero Drum Ltd.[4] It is an ellipsoid made out of polyurethane with dimensions of 2.8 m × 2.8 m × 1.9 m providing axisymmetric aerodynamic stability. The envelop is filled with up to 8 m$^3$ of balloon grade helium capable of lifting 5 kg of payload although at least 1.5 kg of surplus lift is recommended for stable performance. The
10   polyurethane envelop provides a good seal and results in only up to a maximum 0.5 % volume helium loss per day, enabling the system to be used up to 24 hr before a helium recharge is necessary. The tethered navigation system enables deployment of the balloon at a location of interest while controlling the ascend or descend rate. A close-up of the TANAB system during sampling and a system schematic can be seen in Fig. 1.

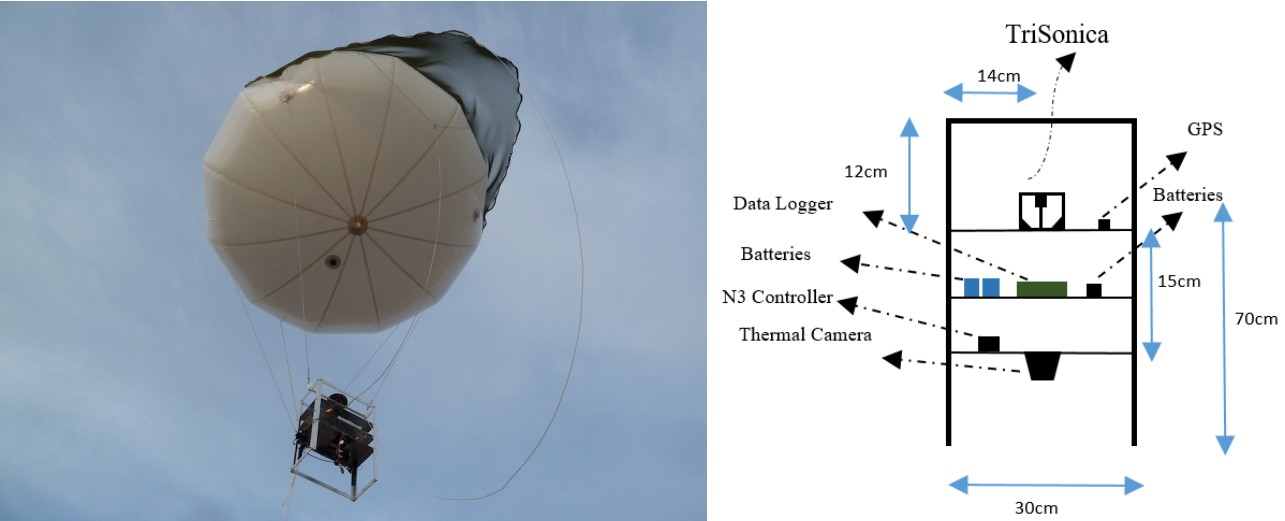

**Figure 1.** Left: close-up of the TANAB system during sampling; right: schematic of the TANAB gondola with all the sensors.

---

[4]https://www.rc-zeppelin.com/



## 2.2 Mini Weather Station

The TriSonica[TM] Mini weather station is an ultrasonic anemometer manufactured by Anemoment[TM] and is mounted onto the gondola of TANAB[5]. This mini weather station is ideal for applications that require a miniature, lightweight, and low velocity anemometer, and are suitable particularly for airborne systems. It has a measurement path length of 35 mm and a weight of

50 gr. The light weight makes it an ideal candidate to use with the TANAB system. It can measure the 3D wind speed, air temperature, relative humidity, and the barometric pressure at a sampling rate up to 10 Hz. The open path provides the least possible distortion of the wind field. Its design with four measurement pathways provides a redundant measurement and the path with the most distortion is removed from the calculations to provide accurate wind measurements. It is also equipped with a compass and a tilt sensor. Because of its low power consumption (only 30 mA at 12 V), it is highly power efficient and can

record data for hours.

A data logger by Applied Technologies Inc.[6] is used as the data synchronization and data collection device for the TriSonica[TM] Mini weather station. This data logger records the measurements on an SD card onboard that can be retrieved after every flight. In addition, the TriSonica[TM] Mini can be monitored or programmed using serial communication via the data logger. The TriSonica[TM] Mini weather station and the data logger are powered using a 12 V Lithium-Polymer (LiPo) battery.

The anemometer measures wind speed in the range $0 - 30 \text{ m s}^{-1}$ at a resolution of $0.1 \text{ m s}^{-1}$. The accuracy of the measurement is $\pm 0.1 \text{ m s}^{-1}$ ($0 - 15 \text{ m s}^{-1}$) or $\pm 2 \%$ ($15 - 30 \text{ m s}^{-1}$). Wind direction is measured at a resolution of $1\,°$ and an accuracy of $\pm 1\,°$. Vertical winds are measured appropriately if the approach elevation angle is within $\pm 30\,°$, a condition that is typically met under calm wind conditions in smoothly varying topography. Temperature is measured in a range from $-25\,°C$ to $+80\,°C$ with a resolution of $0.1\,°C$ and an accuracy of $\pm 2\,°C$. Pressure is measured in the range $50 - 115 \text{ Pa}$ with an accuracy of $\pm 1$

kPa. The tilt sensor measures the pitch and roll with an accuracy of $\pm 0.5\,°$. The compass measures the magnetic heading with an accuracy of $\pm 5\,°$.

## 2.3 Thermal Camera and Flight Controller

The uncooled thermal camera used is the Zenmuse XT, 19-mm lens, manufactured by FLIR Systems[7]. This radiometric camera has a resolution of $640 \times 512$, and it is capable of capturing thermal images at 30 Hz. This camera is powered by DJI and

mounted underneath the gondola of the TANAB using a gimbal kit. The camera is operated by the DJI N3 flight controller[8] with associated hardware securely attached to the gondola of the TANAB. This aerial imaging device is designed to be mounted and operated using drone aircraft systems while it is the first time such a camera has been considered for a balloon system. The camera gimbal system compensates for gondola movement due to wind to keep the camera orientation. The camera system roll angle is generally very close to zero due to the self-stabilization of the device. Nevertheless, the gondola itself is connected

to the balloon using ropes and a swivel mechanism to hang freely without being influenced by aerodynamic vibrations of

---

[5]https://www.anemoment.com/
[6]http://www.apptech.com/
[7]https://www.flir.com/
[8]https://www.dji.com/



the balloon envelop. The gondola is designed to include a safety feature to protect the camera from impact damage. The DJI Lightbridge 2 (LB2) controller is used with either an iOS or Android device to communicate with the N3 and Zenmuse XT during flight. The LB2 has a maximum communicable range for image transmission of 5 km. Pictures and videos can be recorded with precision while the gondola is in flight as the camera can move independently with respect to the gondola.

During flight, the N3 and DJI Zenmuse XT function in parallel such that the GPS location and altitude of the gondola as well as tilt and heading angles of the camera are recorded by the N3 and are included in the meta-data of each image. All GPS information and associated gondola parameters are recorded and stored within the N3 controller and can be retrieved for later analysis using a mini USB connection. The images taken by the thermal camera are saved to an onboard micro SD card. The thermal camera can record images in four different file types including JPEG, R-JPEG, TIFF T-Linear Low, and TIFF T-Linear

High. Using the recorded information for each image, the GPS coordinates (latitude and longitude) for individual pixels within each image can be derived. Through using additional software, surface temperature from individual pixels are calculated.

### 2.3.1  Image Processing Methodology

The images obtained from field observations were processed utilizing Python (version 3.6), ExifTool (version 10.94), ImageMagick (version 7.07), and mathematical relationships to calculate Land Surface Temperature (LST) in Kelvin from pixels

for each image. Furthermore, mathematical and trigonometric relationships were employed to directly georeference the image pixels to the World Geodetic System 1984 (WGS84) by deriving decimal degree latitude and longitude values. ExifTool is a software package used to read, write, and edit metadata from images. Within ExifTool, different tags are used depending on the camera manufacturer to extract relevant metadata. Using the FLIR tag in ExifTool, important metadata from each image was derived from the onboard airborne flight controller. ExifTool is executed through the Linux terminal window. ImageMag-

ick is a software used to edit and create images. When extracting the raw data signal value recorded by the thermal camera, ImageMagick is used in conjunction with the ExifTool tag function. ExifTool specifies the data to return from each image and ImageMagick specifies the exact pixel to extract from the image.

Through using ExifTool, the following metadata parameters from each image are obtained: latitude of camera gimbal, longitude of camera gimbal, camera gimbal roll degree, camera gimbal yaw degree, camera gimbal pitch degree, gondola

roll degree, gondola yaw degree, gondola pitch degree, Planck constant $R_1$, Planck constant $R_2$, Planck constant $B$, offset Planck constant $O$, Planck constant $F$, date image was recorded, altitude of the gondola (only if the TriSonica™ was not operational when the image was captured), the raw signal value recorded by the thermal camera, and the reflected apparent temperature. When airborne, the thermal camera is stabilized, as a result, the camera system is independent of the gondola up to the camera's mechanical range. As a result, images beyond the mechanical extent of the camera were omitted (roll greater

than 45 ° or less than −45 ° and pitch greater than 45 ° and less than −135 ° where the recorded pitch angle is located at the center of the image, as measured from the horizontal plane). Positive pitch angles primarily include the sky and negative pitch angles primarily include the Earth's surface. Furthermore, camera gimbal pitch angles greater than −2 ° were removed from the image processing technique as these images were very oblique and would have contributed to LST errors. Camera gimbal



pitch angles greater than $-30\,°$ are known to introduce possible errors into the LST calculation [9]. Since the TANAB was flown up to a maximum of $150\,\mathrm{m}$ above grade level, only some images greater than $-30\,°$ were omitted from the image analysis as a compromise between LST spatial distribution and LST accuracy. Additionally, any images with incorrect latitude or longitude values were omitted. For georeferencing simplicity, images recorded with a camera gimbal pitch less than or equal to $-76\,°$

were omitted to ensure that the pitch angle for the bottom of each image was greater than $-90\,°$. As per the thermal camera specifications, the vertical field of view of the camera is $26\,°$ and the pitch angle for the bottom of an image is equivalent to the camera gimbal pitch angle minus one half of the vertical field of view. This condition was included to avoid negative horizontal distances with respect to the camera/gondola. The altitude of the gondola and camera were derived from atmospheric pressures recorded by the TriSonica$^{\mathrm{TM}}$ according to the pressure height equation (Eqn. 12). Timestamps from each image and

the TriSonica$^{\mathrm{TM}}$ data were compared to determine the altitude of the gondola when each image was recorded. The pixel row for images that correspond to sky are calculated such that pixels including sky are omitted.

With georeferencing completed, LST values were derived for every 64th pixel across each row. Pixel rows were selected based on a geometric step function such that the majority of the calculated LST coordinates were located near the top of each image. This was method was chosen as the pixels closer to the top of the image would cover more land surface area.

LST was calculated after direct georeferencing using a modified form of Planck's law (described by Çengel and Ghajar (2015) in Eqn. 1) as developed by Martiny et al. (1996) in Eqn. 2

$$E_{b\lambda} = \frac{C_1}{\lambda^5[\exp\left(\frac{C_2}{\lambda T}\right) - 1]}, \tag{1}$$

$$I = \frac{R}{\exp\left(\frac{B}{T}\right) - 1}, \tag{2}$$

where $E_{b\lambda}$ represents the spectral blackbody emissive power, $\lambda$ represents the wavelength of the emitted energy, $T$ represents the skin temperature of the blackbody, $C_1$ and $C_2$ represent numerical constants, $I$ represents the emitted radiation from the imaged surface, $T$ represents the skin temperature of the imaged surface and $R$ represents numerical constants that are dependent on physical camera parameters as determined by the manufacturer during calibration (Martiny et al., 1996).

The raw signal values recorded by the camera are equivalent to Eqn. 3 as described by FLIR-Systems (2010)

$$U_{obj} = \frac{1}{\epsilon}U_{tot} - \frac{1-\epsilon}{\epsilon}U_{refl} - \frac{1-\tau}{\epsilon\tau}U_{atm}, \tag{3}$$

where $U_{obj}$ is the raw output voltage of a blackbody recorded by the thermal camera, $U_{tot}$ is the total raw output voltage recorded by the thermal camera, $U_{refl}$ is the theoretical raw camera output voltage for a blackbody based on the assumed reflective temperature and $U_{atm}$ is the theoretical raw output voltage of a blackbody based on the assumed atmospheric temperature. $\epsilon$ is the emissivity of the blackbody and $\tau$ is the atmospheric transmissivity. The transmissivity of the atmosphere

is generally close to 1.0 (Usamentiaga et al., 2014) under clear weather conditions. From the camera metadata, the assumed

---

[9]https://dl.djicdn.com/downloads/zenmuse_xt/en/sUAS_Radiometry_Technical_Note.pdf





reflective temperature was 22 °C. This value was extracted using ExifTool. The $U_{refl}$ value was calculated using (Zeise and Wagner, 2016)

$$U_{refl} = \frac{R_1}{R_2 \exp(\frac{B}{T_{refl}}) - F} - O, \tag{4}$$

where $R_1$, $R_2$, $B$, $F$ and $O$ are Planck constants of the camera extracted through ExifTool as detailed above.

The Emissivity of the land surface was determined to be a function of geographic position. The Moderate Resolution Imaging Spectroradiometer (MODIS) instrument MOD11B3 data product was used to derive Land Surface Emissivity (LSE) (Wan et al., 2015). The monthly data product with a resolution of 6 km was used to derive LSE for data collected during the observation campaign in May 2018. The emissivity values were calculated using bands 29, 31, and 32 values. Since the thermal camera used a Longwave Infrared Radiation (LWIR) detector, radiation within the 7.5 μm to 13.5 μm spectral range was included in

the camera voltage output[10]. MODIS band 29 records radiation within the 8.4 μm to 8.7 μm spectral range, band 31 records radiation within the 10.78 μm to 11.28 μm spectral range, and band 32 records radiation within the 11.77 μm to 12.27 μm spectral range. These three bands are used in conjunction with the Broad Band Emissivity (BBE) derivation to calculate emissivity as a function of geographical area (Wang et al., 2005). The BBE formula used is

$$BBE = a\epsilon_{29} + b\epsilon_{31} + c\epsilon_{32}, \tag{5}$$

where $a$, $b$, and $c$ are constants that vary based on the land surface material. Wang et al. (2005) determined that the constants $a$, $b$, and $c$ do not vary significantly between soil, vegetation, or anthropogenic materials. However, water ice and snow resulted in noticeably different BBE coefficients. Based on Wang et al. (2005), the BBE coefficients for $a$, $b$, and $c$ were chosen to be 0.2122, 0.3859, and 0.4029, respectively.

The output voltage value for the object was then calculated. Finally, the LST value as per the thermal camera was derived

using

$$T_{obj} = \frac{B}{\ln(\frac{R_1}{(R_2(U_{obj}+O)} + F)}. \tag{6}$$

## 3   Calibration Experiments

In order to check the validity of the high frequency data from the anemometer onboard of TANAB, the anemometer performance was validated against calibrated sensors prior to the field campaign in a series of wind tunnel and outdoor calibration

experiments.

### 3.1   Wind Velocity Calibration

The mounted anemometer performance was characterized in a highly turbulent flow generated by a wind tunnel at the University of Guelph. The YOUNG 81000 ultrasonic anemometer, which is already calibrated, is used for cross comparison to

---

[10]https://www.dji.com/zenmuse-xt/info/





derive the calibration coefficients for the anemometer using line fits. Both sensors were set up at similar airflow condition while wind speed was varied at three or ten different wind speed levels in the range $2 - 10 \ \mathrm{m \, s^{-1}}$. At each wind speed level, data recording continued for $5 \ \mathrm{min}$. Each recording was time averaged to calculate mean and turbulence statistics. It is customary to use notation associated with Reynolds averaging to decompose a signal $X$ into mean $\overline{X}$ and turbulent $x$ components such that $X = \overline{X} + x$. The calibration factors were derived for the $x$, $y$, and $z$ components of mean velocity, i.e. $\overline{U}$, $\overline{V}$ and $\overline{W}$, in separate experiments. The calibration equations are given below. While Trisonica™ underpredicts horizontal wind velocities ($\overline{U}$ and $\overline{V}$), it overpredicts the vertical wind velocity ($\overline{W}$), all of which can be corrected by the calibration fits

$$
\begin{aligned}
\overline{U}_{YOUNG} &= 0.85 \times \overline{U}_{TriSonica} + 0.13, \\
\overline{V}_{YOUNG} &= 0.81 \times \overline{V}_{TriSonica} + 0.75, \\
\overline{W}_{YOUNG} &= 2.80 \times \overline{W}_{TriSonica} - 0.12.
\end{aligned}
\tag{7}
$$

### 3.2 Temperature Calibration

Temperature measured by TriSonica™ is calibrated with respect to the Campbell Scientific HMP60 sensor[11]. The latter collected minute-averaged temperatures, to which the TriSonica™ temperatures were also averaged and compared. The experiment was carried out under a set of climate conditions to cover a wide range of temperatures outdoors. The calibration equation obtained is given below. As a result of this multipoint calibration, TriSonica™ has underpredicted temperature, which can be corrected.

$$
\overline{T}_{HMP60} = 1.24 \times \overline{T}_{TriSonica} - 73.89
\tag{8}
$$

### 4 Field Experiments and Results

The TANAB system was launched at a mine facility in northern Canada (above $56 \ °\mathrm{N}$) for an environmental monitoring field campaign in May 2018. A schematic of the mine facility can be seen in Fig. 2. The depth of the mine is approximately $100 \ \mathrm{m}$.

TANAB flew for more than $50 \ \mathrm{hr}$ collecting data. The objectives of the measurements were to determine dynamics of the atmosphere at different diurnal times (e.g. day versus night) and locations (near tailings pond versus inside the mine). Such dynamics determine the transport of green house gases (GHGs) and therefore emission fluxes. Measurements of the GHG fluxes were not the objective of this paper and will be addressed elsewhere.

Surface level transport mechanisms strongly depend on atmospheric dynamics. Factors such as wind speed, atmospheric diffusion coefficient, and thermal stability greatly influence emission fluxes. As a result, the particular focus of this study is measurement of surface level meteorology in the lowest $200 \ \mathrm{m}$ altitudes and earth surface temperature. The launch details are summarized in Table 1.

[11]https://www.campbellsci.com/



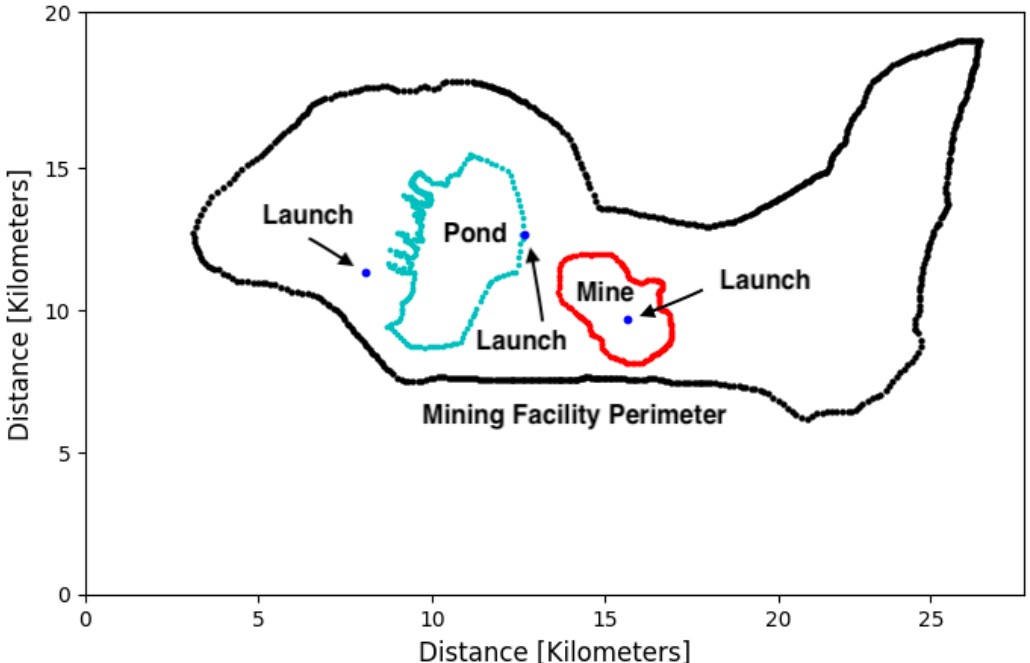

**Figure 2.** A schematic of the mine facility. The black dots represent the outline of the entire facility. The green dots represent the outline of the tailings pond, and the red dots represent the outline of the mine. The blue dots are the balloon launch locations.

Vertical transport of momentum and heat are predominant processes within the the surface layer of the ABL (Businger et al., 1971) and deriving the vertical fluxes of momentum and heat from wind speed and temperature profile measurements can be achieved using TANAB. Turbulence kinetic energy is one of the key measures of turbulence in the atmosphere as it controls the vertical and horizontal mixing (Lenschow et al., 1980; Svensson et al., 2011; Shin et al., 2013; Canut et al., 2016). It is also used for the parameterization of small-scale turbulent transport processes, such as vertical fluxes, when the smaller scale motions are not modelled directly (Aliabadi et al., 2018b). The equation that represents turbulence kinetic energy is

$$k = \frac{1}{2}(\overline{u^2} + \overline{v^2} + \overline{w^2}), \tag{9}$$

where $\overline{u^2}$, $\overline{v^2}$, and $\overline{w^2}$ are variances of turbulent velocity fluctuations along the $x$, $y$, and $z$ directions. Another coordinate system to analyze mean wind velocity components and wind velocity fluctuations is the *along-wind* coordinate system, in which the horizontal axis is rotated toward the direction for mean horizontal wind velocity vector (Aliabadi et al., 2016c). In this system mean horizontal wind speed is defined as

$$\overline{S} = \sqrt{\overline{U}^2 + \overline{V}^2}, \tag{10}$$





**Table 1.** Tethered And Navigated Air Blimp (TANAB) Launch details

| Experiment | Location | Start Date | Start time | End time | No. of profiles | Experiment time |
|---|---|---|---|---|---|---|
| 1 | Tailings pond | 2018:05:07 | 21:41:00 | 02:47:00 | 14 | 05:06:00 |
| 2 | Tailings pond | 2018:05:09 | 03:30:00 | 04:00:00 | 02 | 00:30:00 |
| 3 | Tailings pond | 2018:05:10 | 02:30:00 | 08:30:00 | 21 | 06:00:00 |
| 4 | Tailings pond | 2018:05:15 | 04:55:00 | 11:00:00 | 22 | 06:05:00 |
| 5 | Mine | 2018:05:18 | 04:12:00 | 11:12:00 | 20 | 07:00:00 |
| 6 | Mine | 2018:05:19 | 18:52:00 | 23:15:00 | 17 | 04:23:00 |
| 7 | Mine | 2018:05:21 | 11:00:00 | 12:17:00 | 04 | 01:17:00 |
| 8 | Mine | 2018:05:23 | 01:47:00 | 05:30:00 | 10 | 02:43:00 |
| 9 | Mine | 2018:05:24 | 11:19:00 | 14:25:00 | 12 | 03:06:00 |
| 10 | Mine | 2018:05:27 | 14:38:00 | 17:50:00 | 18 | 03:12:00 |
| 11 | Tailings pond | 2018:05:30 | 10:55:00 | 18:57:00 | 24 | 08:02:00 |
| 12 | Tailings pond | 2018:05:31 | 11:07:00 | 14:43:00 | 08 | 03:36:00 |

where $\overline{U}$ and $\overline{V}$ are mean wind velocity vectors along the conventional $x$ and $y$ coordinate axes. Still the velocity fluctuations along wind can be considered using the Reynolds decomposition $S = \overline{S} + s$. With this coordinate system the turbulence kinetic energy simplifies to

$$k = \frac{1}{2}(\overline{s^2} + \overline{w^2}). \tag{11}$$

We used the pressure altitude as the most convenient proxy for actual altitude based on the national weather service formula[12]

$$z = 0.3048 \left( 1 - \left( \frac{p}{1013} \right)^{0.19} \right) \times 1.45 \times 10^5, \tag{12}$$

where pressure measurement $p$ is in $\mathrm{mBar}$ and pressure altitude $z$ is in $\mathrm{m}$.

### 4.1  Sampling Time

TANAB measured wind speed, the turbulence kinetic energy, variances, and fluxes for both momentum and heat rigorously. Each balloon launch lasted approximately $15-30\,\mathrm{min}$ while the tether was carefully controlled to obtain a profile with constant ascent and descent rates. The sampling time for calculating the mean and turbulence quantities were $3\,\mathrm{min}$ while typically it is 10 to $30\,\mathrm{min}$ for flux tower measurements (Aliabadi et al., 2018a). For turbulence statistics, the time series is first detrended to ensure that background weather variations that have inherently very large time and length scales are filtered out without

influencing the turbulence statistics calculations. It is known that finite time sampling, instead of ensemble averaging, will introduce random and systematic errors in the prediction of turbulence statistics such as variances and fluxes (Lenschow et al.,

---

[12]https://www.weather.gov/epz/wxcalc_pressurealtitude





1994). While random errors could result in overprediction or underprediction of turbulence statistics, the systematic errors always underpredict the magnitude of the turbulence statistic. These errors have been reported in an aircraft campaign to be anywhere in the range 10 to 90 % of the measured value (Aliabadi et al., 2016b). While repeated measurements, increasing the averaging time, and detailed error analysis are possible to eliminate such errors from predictions, the focus of this study was not

to investigate errors associated with finite sampling time. Nevertheless, operating the TANAB involves a delicate choice of the sampling time. On one hand, short sampling times have inherent large errors but provide profiles at high vertical resolutions. On the other hand, long sampling times have inherent small errors but provide profiles at low vertical resolutions.

## 4.2   Diurnal Variation in Wind Speed and Turbulence Statistics

It was observed that both wind speed (Fig. 3) and turbulence kinetic energy (Fig. 4) exhibited a significant diurnal variation,

indicating calm conditions at nights and early mornings, when atmospheric diffusion coefficient is low, and gusty conditions in the mid-afternoons when the atmospheric diffusion coefficient is high. When calculating statistical percentiles, the data is combined over all altitudes and aggregated for both the mine and tailings pond locations.

Similar diurnal variations can be also observed in the case of other turbulence statistics such as vertical momentum flux (or Reynolds stress) $\overline{sw}$, vertical sensible kinematic heat flux $\overline{w\theta}$, potential temperature variance $\overline{\theta^2}$, and vertical velocity

variance $\overline{w^2}$ (Fig. 5). The measurement of weak turbulence in the nocturnal boundary layer is also very important as it leads to weak turbulent dispersion and large accumulation of heat or atmospheric constituents in the lower part of the stable boundary layer (Mahrt and Vickers, 2003, 2006). Periods of strong stability with intermittent turbulence can also occur in the nocturnal boundary layer (Businger et al., 1971; Mahrt, 1999).

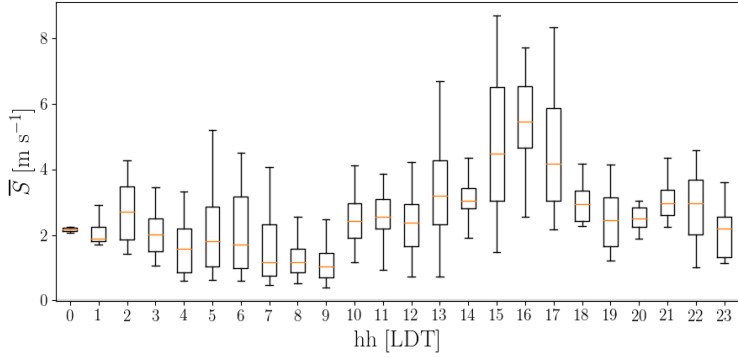

**Figure 3.** Diurnal variation of wind speed. At each hour observations are plotted using statistical percentiles (5th, 25th, 50th, 75th, and 95th); Local Daylight Time (LDT).





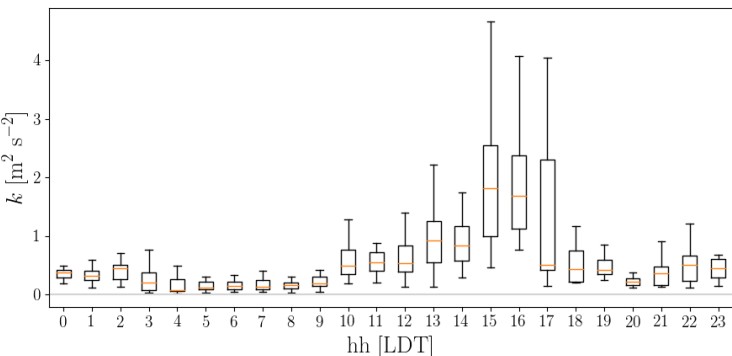

**Figure 4.** Diurnal variation of turbulence kinetic energy. At each hour observations are plotted using statistical percentiles (5th, 25th, 50th, 75th, and 95th); Local Daylight Time (LDT).

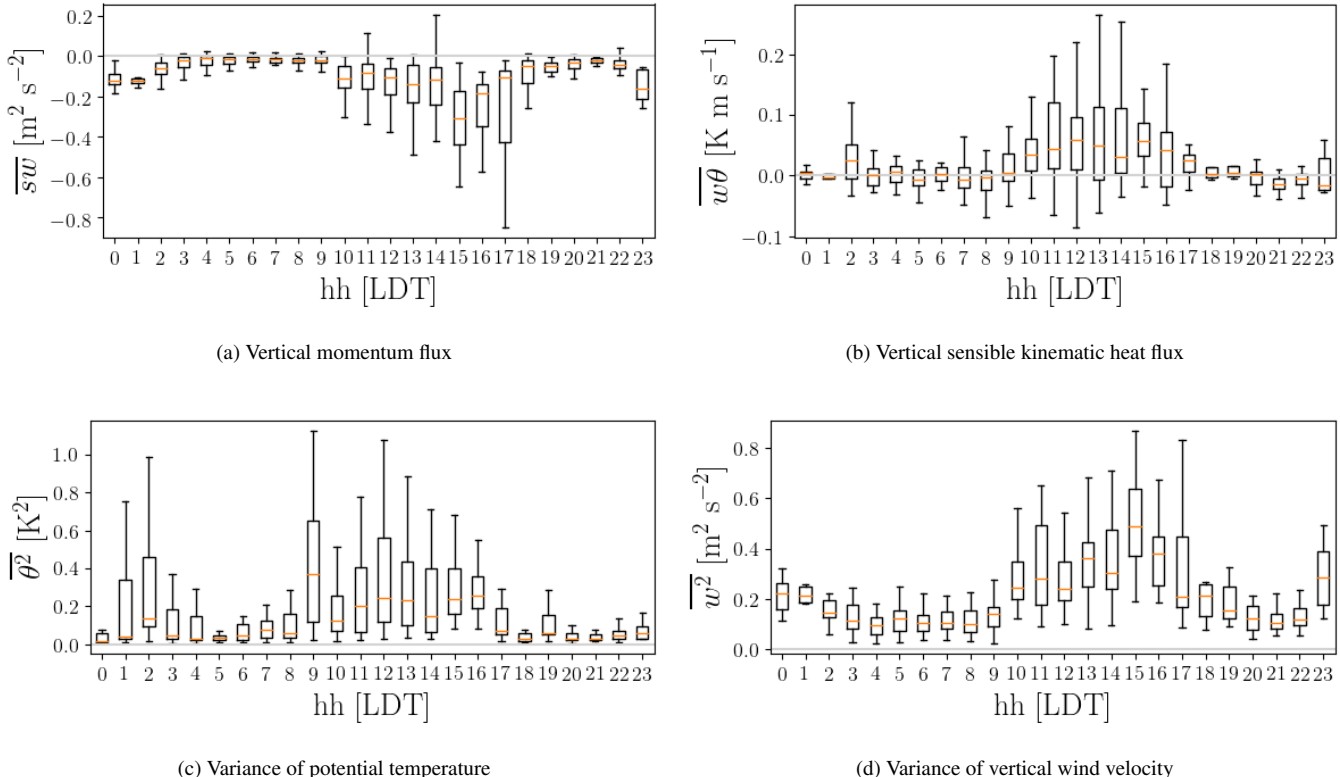

(a) Vertical momentum flux

(b) Vertical sensible kinematic heat flux

(c) Variance of potential temperature

(d) Variance of vertical wind velocity

**Figure 5.** Diurnal variation of different turbulence statistics. At each hour observations are plotted using statistical percentiles (5th, 25th, 50th, 75th, and 95th); Local Daylight Time (LDT).



## 4.3 Vertical Variation of Mean and Turbulence Statistics

Figure 6 shows the vertical profiles of turbulence statistics for different diurnal periods (4-hr intervals) of the day. 50th percentiles are shown for all observations. Data is binned in 20-m height intervals. The vertical structure of the atmosphere near the surface can be understood while analyzing the turbulence kinetic energy $k$, variance of along-wind horizontal wind velocity
$\overline{s^2}$, vertical momentum flux $\overline{sw}$, vertical sensible kinematic heat flux $\overline{w\theta}$, variance of potential temperature $\overline{\theta^2}$, and variance of vertical wind velocity $\overline{w^2}$.

It is noteworthy that maximum flight altitude is usually less under windy conditions, therefore most profiles obtained under windy conditions in mid day are shorter than those obtained under calm conditions at night and early mornings. It is observed that in the surface layer within ABL the highest gradients of turbulence properties occur at the lowest 100 m. This statement
must be considered with caution. Certainly it is only valid for the short length and time scales considered as a result of the 3-min time averaging. The diurnal variation of turbulence statistics are clear from the plots. The magnitude for most statistics are greatest during mid day time interval $1200-1600$ LDT due to gusty conditions, while the magnitudes are smallest during night-time and early-morning time interval $0400-0800$ LDT associated with calm conditions. The vertical momentum flux $\overline{sw}$ is always negative, confirming boundary layer physics that imply momentum should sink to the surface due to skin drag with
turbulent processes. This also implies confidence in the measurement of velocity fluctuation covariance. The vertical sensible kinematic heat flux $\overline{w\theta}$ is negative at night time but positive during day time. This confirms that the earth surface acts as a heat sink at night time due to heat loss by radiation to the sky, and that the earth surface acts as a heat source during day time due to heat gain by radiation from the sky. The variance of potential temperature $\overline{\theta^2}$ is not strictly diminished under calm night time conditions. This can be interpreted as the presence of differential near surface horizontal gradients of temperature due to
thermal structures. It is expected that such gradients must exist because of the heterogeneity of land surface and anthropogenic activities in such a complex terrain of a mining facility.

## 4.4 Variation of Thermal Stability and Wind Speed as a Function of Diurnal Time

The atmospheric dynamical condition can be described using two parameters: 1) wind speed and 2) thermal stability. The wind speed determines mechanical advection and usually the higher the wind speed the greater the atmospheric transport and
diffusion coefficient. Thermal stability determines the buoyant transport in the vertical direction in the atmosphere. Thermal stability is reported using various methods such as i) the vertical gradients of the potential temperature (Liu and Liang, 2010), ii) the bulk Richardson number (Mahrt, 1981; Aliabadi et al., 2016a), or iii) Monin-Obukhov length (Obukhov, 1971; Wilson, 2008).

If vertical gradients of the potential temperature or the bulk Richardson number are positive, the atmosphere is stable and
the buoyant transport is suppressed. This occurs during the nights and early mornings. If the vertical gradients of the potential temperature and bulk Richardson number are negative, the atmosphere is unstable and buoyant transport is enhanced. This occurs during the mid afternoons. If the vertical gradients of potential temperature and bulk Richardson number are close to zero, the atmosphere is neutral, in which case buoyant transport is still present but weak.

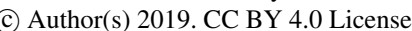



**Figure 6.** Vertical profiles of different turbulence statistics for different diurnal time periods (4-hr intervals) of the day; times in Local Daylight Time (LDT).



Figures 7 and 8 show evidence for the variation of vertical profiles of thermal stability and wind speed as a function of diurnal time, respectively. Here the data is statistically processed in 3-min intervals, such that a median is calculated for each 3-min interval. Since observations for these plots are not aggregated over many days, the height interval is not binned. The thermal stability and magnitude of wind speed differences between day and night are clearly evident. However, the profiles of wind speed do not exhibit the expected power law or logarithmic law as a function of height. This is likely due to limited time of sampling.

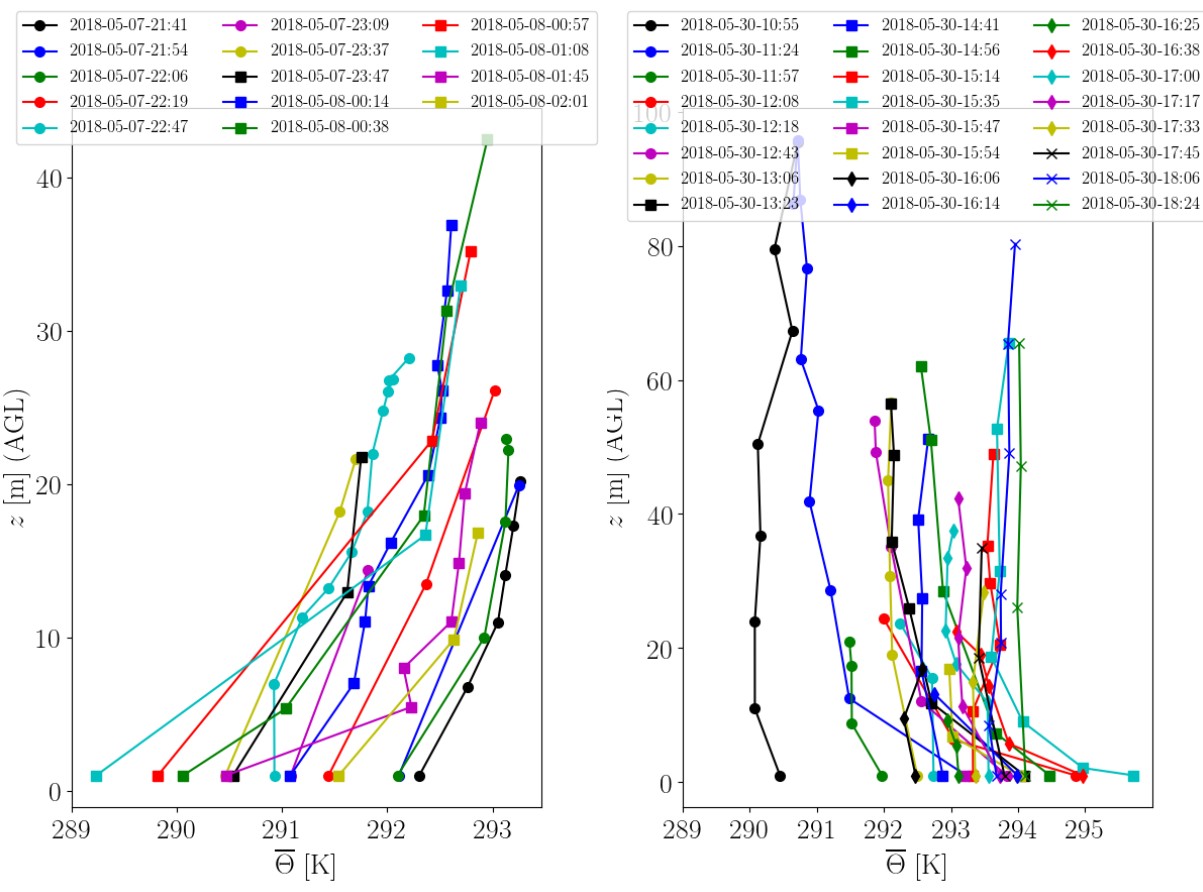

**Figure 7.** Vertical profiles of potential temperature on 7 May 2018 and 30 May 2018; Left: thermally stable condition at night and early morning when the gradients are positive near the surface; Right: thermally unstable condition at mid day when the gradients are negative near the surface.





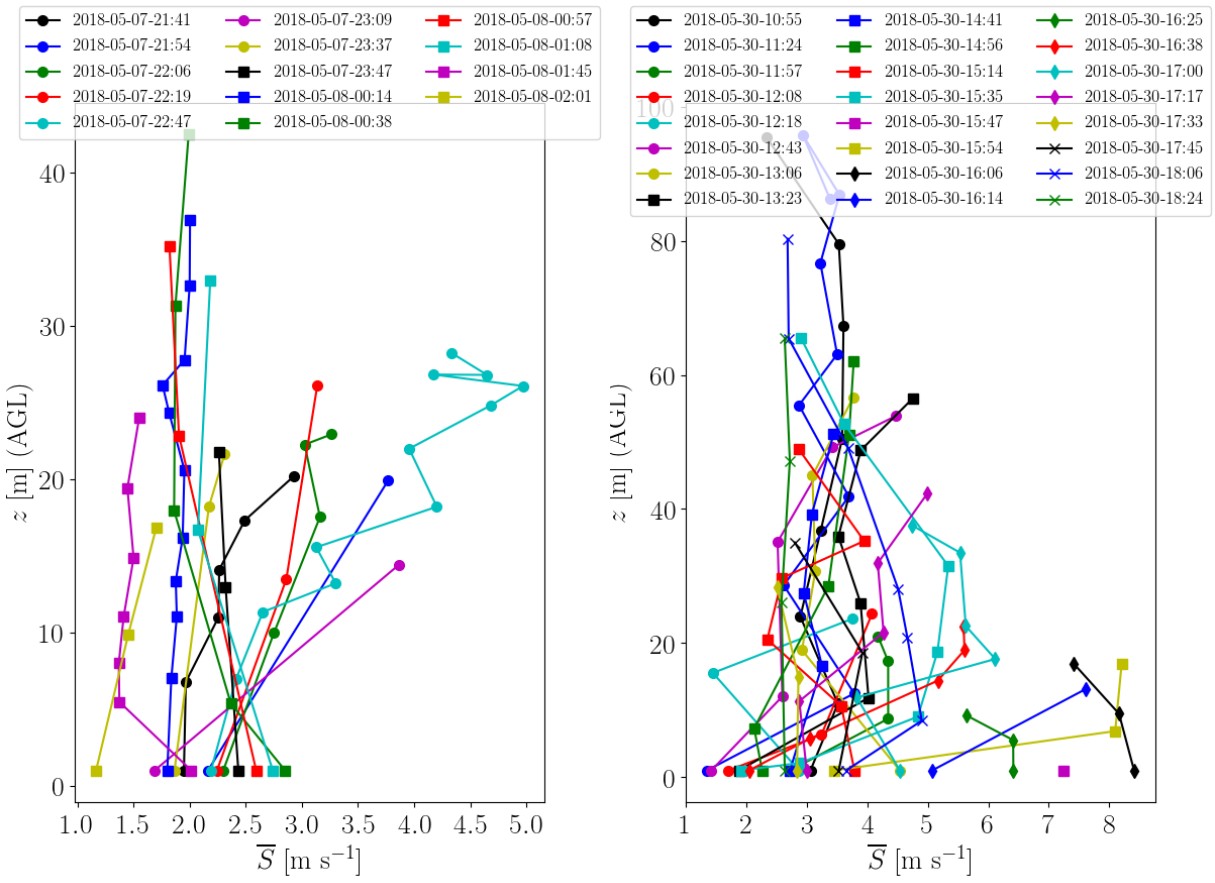

**Figure 8.** Vertical profiles of wind speed on 7 May 2018 and 30 May 2018. Left: thermally stable condition at night and early morning, Right: thermally unstable condition at midday.

## 4.5 Atmospheric Dynamical Condition

As shown in Fig. 9, all observations were used to determine the atmospheric dynamical condition on a two-dimensional map consisting of thermal stability, i.e. bulk Richardson number $Ri_b$, and mean wind speed $\overline{S}$. Here the bulk Richardson number is defined as

$$Ri_b = \frac{gH}{(\overline{S}_H - \overline{S}_S)^2} \frac{\overline{\Theta}_H - \overline{\Theta}_S}{\overline{\Theta}_A}, \tag{13}$$

where $g$ is gravitational acceleration, $H$ is the maximum altitude for each launch, $\overline{S}_H$ is mean horizontal wind speed at maximum altitude, $\overline{S}_S$ is mean horizontal wind speed near the surface, $\overline{\Theta}_H$ is mean potential temperature at maximum altitude, $\overline{\Theta}_S$ is mean potential temperature near the surface, and $\overline{\Theta}_A$ is mean potential temperature for the entire launch.




The frequency plot shows the most frequent status of the atmosphere by providing a normalized count of each pair of observed $Ri_b$ and $\overline{S}$, while the colour plot shows the median value for turbulence kinetic energy $k$ given as a function of each pair of observed $Ri_b$ and $\overline{S}$. It is found that the atmosphere spends a considerable amount of time under near-neutral and stable condition at nights and early mornings where $Ri_b \geq 0$, possibly with the same likelihood of unstable state during mid day

where $Ri_b < 0$. The colour plot for turbulence kinetic energy shows that the highest values are observed under near-neutral conditions where $Ri_b \sim 0$ and mean wind speed is high such that $\overline{S} > 5\,\mathrm{m\,s^{-1}}$. A few spurious high frequency of observations are detected for very large Richardson numbers $Ri_b \sim 10$ and very low wind speeds $\overline{S} \sim 1-2\,\mathrm{m\,s^{-1}}$. These are likely due to the inability of TANAB to detect mean wind speed gradients under calm conditions. These plots demonstrate that the atmosphere in the surface layer has *preferred* states. For instance, it was never observed to be very stable and gusty at the same time.

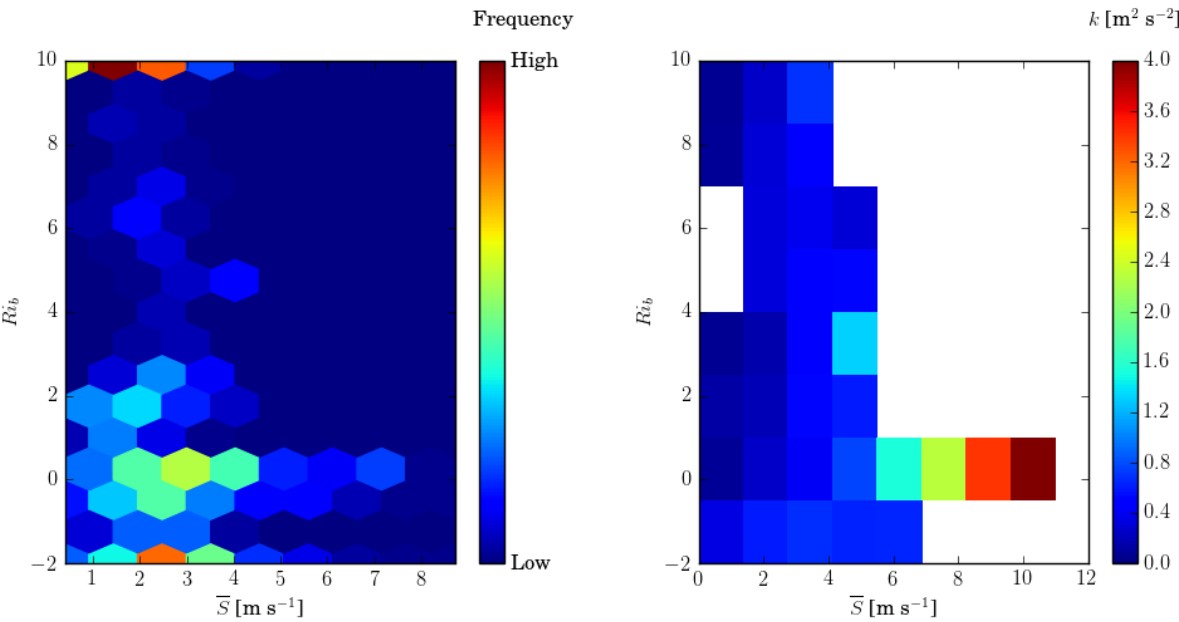

**Figure 9.** Left: frequency plot of atmospheric dynamical condition as a function of bulk Richardson number $Ri_b$ and mean wind speed $\overline{S}$; Right: turbulence kinetic energy $k$ as a function of the same parameters.

**4.6   Comparison Between the Mine and the Tailings Pond**

Almost the same number of hours were spend measuring the surface layer at the mine and near the tailings pond using TANAB. The measurements attempted to cover a 24-hr time period in three to four days so that it would capture the diurnal variations completely. Note that due to logistical difficulties, it was impossible to measure the surface layer in either location for 24 hr continuously. The base location for the launches at the mine was approximately 100 m below grade and at the center of the

mine, while the base location near the pond was on the east side. We have compared the diurnal variations in various surface layer properties between the mine and near the pond observations. These include the mean horizontal wind speed, turbulence



kinetic energy, vertical momentum flux, vertical sensible kinematic heat flux, variance of vertical velocity vector, and variance of potential temperature. We have also compared the atmospheric dynamical condition in terms of the bulk Richardson number and mean wind speed. The idea behind this comparison was to quantify any differences in surface layer properties as a result of terrain complexity and land use.

### 4.6.1 Comparison of Diurnal Variation of Turbulence Properties

TANAB captured the diurnal variation of mean and turbulence statistics in the mine as well as near the tailings pond. In Fig. 10 we plot the diurnal variations for the mine in red and near the tailings pond in blue. It is observed that, for both in the mine and near the tailings pond, all the turbulence parameters exhibit a significant diurnal variation, indicating calm conditions at nights and early mornings, when atmospheric diffusion coefficient is low, and gusty conditions in the mid afternoons when the atmospheric diffusion coefficient is high. The data is aggregated for all altitudes.

There are subtle differences between observations at the mine and near the tailings pond. The trends in mean wind speed $\overline{S}$ and turbulence kinetic energy $k$ suggest that the mine experiences calm conditions at early morning hours, i.e. from 0500 to 0700 LDT, when the atmospheric stability is at its maximum, while at the same time near the pond higher mean wind speeds and turbulence kinetic energy are observed. During early afternoons and under unstable conditions, i.e. from 1500 to 1700 LDT, near the pond higher mean wind speeds and turbulence kinetic energy are observed in comparison to the mine. Even though day to day variations of meteorological conditions may contribute to this, there is evidence that such features can also result from terrain complexity and land surface heterogeneity. As far as the vertical sensible kinematic heat flux $\overline{w\theta}$ and variance of potential temperature $\overline{\theta^2}$ are concerned, the mine shows significantly higher activities compared to the location near the pond. In fact, the mine shows significantly positive vertical heat flux and substantial potential temperature variance from 0200 to 0300 LDT. It is known that excavation activities using heavy machinery is very active in the mine over night, possibly explaining the significance of heat flux and potential temperature variance.

### 4.6.2 Comparison of Atmospheric Dynamical Condition

Figure 11 shows the frequency plots of atmospheric dynamical condition as a function of the bulk Richardson number $Ri_b$ and mean wind speed $\overline{S}$ in the mine and near the tailings pond separately. It was found that in the case of tailings pond, the atmosphere spends a considerable amount of time under near-neutral and unstable conditions in comparison to the mine. Again the frequency plots should be read with care at $Ri_b \sim 10$ due to the lack of a reliable TANAB predictions for vertical mean wind speed gradients under calm conditions.

### 4.7 Earth Surface Thermal Imaging

An important measurement of the TANAB system was the earth surface thermal imaging. The surface temperature values and thermal gradients at the mining site are important physical parameters that can potentially drive unique airflow patterns. For instance, hot spots can create rising thermal structures in the boundary layer that subsequently carry surface-level atmospheric





(a) Diurnal variation of mean wind speed

(b) Diurnal variation of turbulence kinetic energy

(c) Vertical momentum flux

(d) Vertical sensible kinematic heat flux

(e) Variance of potential temperature

(f) Variance of vertical wind velocity

**Figure 10.** Diurnal variation of different mean and turbulence statistics; at each hour observations are plotted using statistical percentiles (5th, 25th, 50th, 75th, and 95th); Local Daylight Time (LDT).





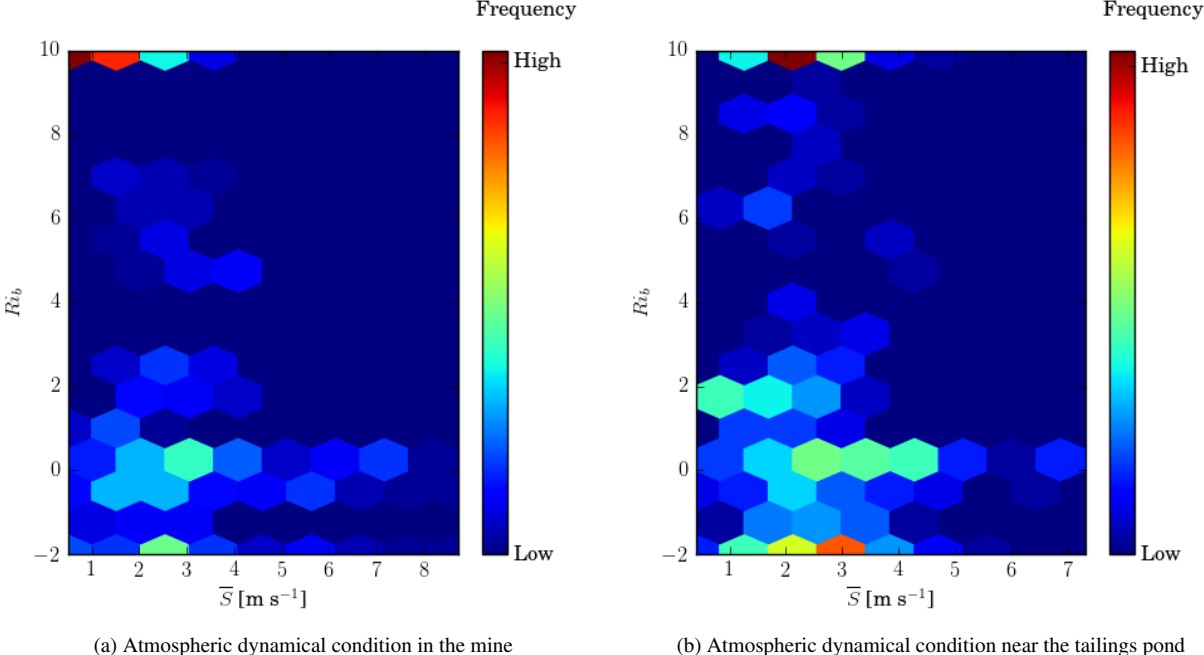

(a) Atmospheric dynamical condition in the mine    (b) Atmospheric dynamical condition near the tailings pond

**Figure 11.** Frequency plots of atmospheric dynamical condition as a function of bulk Richardson number $Ri_b$ and mean wind speed $\overline{S}$ in the mine (left) and near the tailings pond (right).

constituents, such as GHGs, upward. On the other hand, cold spots can create stable boundary layers and initiate subsiding flow that prevent mixing of gaseous compounds within the boundary layer. Inevitably, the thermal condition places a significant impact on the gaseous emission fluxes. TANAB took more than 10,000 thermal images during the May 2018 campaign, all of which are processed in detail to determine the surface temperature characteristics of the site during different diurnal times and

at different locations. An analysis of selected thermal images is included below.

Figure 12 shows the median temperature maps for the site for different diurnal periods (4-hr intervals) of the day. Data is aggregated over all observations measured by the experiments as mentioned in Table 1. The diurnal variation of the temperature is evident. The surface temperature is the greatest during mid day time interval $1200 - 1600$ LDT, while the magnitudes are smallest during night time and early morning time interval $0400 - 0800$ LDT.

In order to quantify the difference in temperatures as a consequence of terrain heterogeneity and other complexities between the mine and tailings pond, we have compared the surface temperatures using box plots in Fig. 13 at different diurnal periods (4-hr intervals) aggregated over the entire observation period as detailed in Table 1. The thermal imaging data is in agreement with the TriSonica™ data as plotted in Fig. 10. For many periods the mine shows higher activity in heat related turbulence statistics such as the turbulent kinematic vertical heat flux during $0800 - 2000$ and variance of potential temperature during $0000 - 0400$.

These time intervals correspond to periods when the mine shows a statistically significant higher surface temperature than the pond using the box plots.





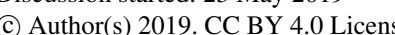

(a) Median temperatures at 0000 − 0400 LDT

(b) Median temperatures at 0400 − 0800 LDT

(c) Median temperatures at 0800 − 1200 LDT

(d) Median temperatures at 1200 − 1600 LDT

(e) Median temperatures at 1600 − 2000 LDT

(f) Median temperatures at 2000 − 2400 LDT

**Figure 12.** Median temperature maps for the site at different periods (4-hr intervals) of the day; median temperature calculated for tiles at a resolution of 1 km × 1 km.


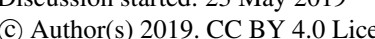

(a) Boxplot of temperatures at $0000 - 0400$ LDT

(b) Boxplot of temperatures at $0400 - 0800$ LDT

(c) Boxplot of temperatures at $0800 - 1200$ LDT

(d) Boxplot of temperatures at $1200 - 1600$ LDT

(e) Boxplot of temperatures at $1600 - 2000$ LDT

(f) Boxplot of temperatures at $2000 - 2400$ LDT

**Figure 13.** Boxplot of temperatures for tailings pond and mine for different periods (4-hr intervals) of the day.





The median temperatures measured by TANAB are compared with satellite measurements of MODIS on 24 May 2018. For this comparison, the selected images for TANAB were captured during $1211 - 1401$ LDT, and for MODIS the selected data source corresponded to 1230 LDT. The horizontal resolution for this comparison was $1 \, \mathrm{km} \times 1 \, \mathrm{km}$. The comparison and the calculated percentage relative error are shown in Fig. 14. The percentage relative error is less than 6 % everywhere within

the perimeter of the mining facility and the median relative error within the mining facility is 1.1%. However, the LST errors are higher in the northwest direction and lower in the southeast direction. These directions are along the topographical height variation, which is highest in the northwest and lowest in the southeast. It appears that TANAB temperature predictions in the northwest direction result from more oblique-angle observations, which are prone to a higher error.

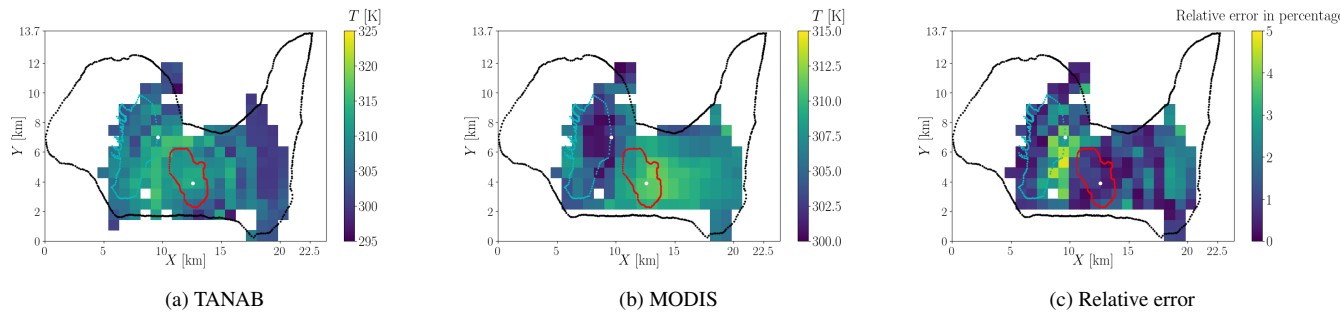

|          (a) TANAB          |          (b) MODIS          |       (c) Relative error       |

**Figure 14.** Comparison of median temperatures measured by TANAB and MODIS observations in the early afternoon on 24 May 2018; TANAB and MODIS temperatures calculated for tiles at a resolution of $1 \, \mathrm{km} \times 1 \, \mathrm{km}$.

As shown by this surface temperature analysis, TANAB offers many advantages over existing satellite systems that attempt

to observe the earth surface temperatures. On the one hand, geostationary satellites, such as the GOES-R, exhibit very low horizontal resolutions ($\sim 1 - 2 \, \mathrm{km}$) and high temporal resolutions (15 minutes) at such high altitudes (Cintineo et al., 2016; Schmit et al., 2017). On the other hand, orbiting satellites capable of thermal imaging at high horizontal resolutions ($\sim 10 - 100$ m), such as the LANDSAT group of satellites, pass over northern sites very intermittently once every sixteen days (Weng, 2012; Chastain et al., 2019). The TANAB overcomes these difficulties by allowing frequent observations at high horizontal

resolutions.

## 5   Conclusions and Future Work

The vertical structure of the Atmospheric Boundary Layer (ABL) in an orographically complex terrain, such as a mine, can be complicated. The Tethered And Navigated Air blimp (TANAB) has provided an acceptable platform for the meteorological measurements inside the surface layer within ABL and atmospheric dynamical condition over a complex terrain of a mining

facility. The field campaign occurred in May 2018 in northern Canada. In the surface layer, most atmospheric transport mechanisms are highly influenced by the terrain geometry. TANAB measured the microclimate in the complex terrain by quantifying mean and turbulence statistics of the atmospheric meteorology. This was achieved by sensing wind speed, wind direction, tem-




perature, relative humidity, and pressure. The calculated parameters included mean horizontal wind speed, turbulence kinetic energy, vertical momentum flux, vertical sensible kinematic heat flux, variance of potential temperature, and variance of vertical wind velocity. In addition, TANAB performed earth surface thermal imaging (temperature mapping) to further characterize how the surface layer interacted with surface temperatures. TANAB further determined the atmospheric dynamical condition
by specifying the combination of the thermal stability state (bulk Richardson number) and mean horizontal wind speed.

TANAB observed that both wind speed and turbulence kinetic energy exhibit a significant diurnal variation, indicating calm conditions at nights and early mornings, when atmospheric diffusion coefficient is low, and gusty conditions in the mid afternoons, when the atmospheric diffusion coefficient is high. TANAB also revealed the vertical structure of the atmosphere near the surface for most meteorological parameters. The highest turbulence kinetic energies occurred in the lowest 100 m
above the surface, albeit for small fluctuation time and length scales probed. Experiments provided evidence for the variation of thermal stability and wind speed as a function of diurnal time. The atmosphere spent a considerable amount of time under near-neutral and stable conditions, with implications on atmospheric diffusion coefficient and emission fluxes of atmospheric constituents released near the surface. The experiments specifically observed differences in the microclimate in the mine pit in comparison to near the tailings pond. The overall pattern of diurnal variation was found to be similar for both the mine and
the tailings pond, but subtle meteorological differences were observed. The mean wind speed and turbulence kinetic energy were comparatively lower in the mine than near the tailings pond in early morning hours, suggesting that the mine boundary layer may have been isolated from the boundary layer above grade. In addition, significantly positive potential temperature variance and turbulent heat fluxes were observed in the mine in various diurnal time windows in comparison to the area near the pond. This was likely due to terrain complexity and anthropogenic activities in the mine. TANAB detected a significant
horizontal temperature difference between the mine and surrounding in comparison to the tailings pond. The mine exhibited greater surface temperatures in many diurnal time periods, likely due to land surface modifications and anthropogenic activities. Higher surface temperatures at the mine could explain the meteorological effects observed in the surface layer of the mine.

A particular challenge in operating the TANAB system is the proper choice of sampling time. On the one hand, short sampling times impose inherent errors in mean and turbulence statistics predictions while enable high resolution vertical
measurements. On the other hand, long sampling times impose less inherent errors in mean and turbulence statistics predictions but they only enable low vertical resolution measurements. Another drawback of TANAB system is that it is not autonomous, so it requires intensive operator effort to fly it. Future development requires advanced techniques for autonomous control of TANAB.

Overall, TANAB offers a simple and cost-effective platform for microclimate measurements within the atmospheric sur-
face layer. The light high-frequency weather sensor onboard enables measurement of mean and turbulence statistics of the atmospheric transport. This configuration allows a wide spatiotemporal coverage compared to fixed flux towers. The thermal imaging system enables high resolution temperature mapping of remote environments frequently, thus overcoming current limitations of satellite systems. TANAB can potentially provide meteorological data as boundary conditions or validation datasets for developing high resolution Computational Fluid Dynamics (CFD) and other mesoscale models that attempt simulating
meteorological processes and emission fluxes from large complex terrains of mining and other similar facilities.




*Code and data availability.* The Atmospheric Innovations Research (AIR) Laboratory at the University of Guelph will make the thermal image processing code and validation data publicly available at www.aaa-scientists.com in the near future. In the meantime, the source code and the supporting environmental field monitoring data can be requested from the Principal Investigator, Amir A. Aliabadi, via the authorization of data owners (aliabadi@uoguelph.ca).

*Author contributions.* The field experimental data was collected by MKN, AN, RB, MRN, MM, and AAA. The analysis of anemometer data was performed by MKN and AAA. The analysis of thermal imaging was performed by RB. Anemometer calibration was performed by AN, MM, and AAA. The funding was acquired by AAA. Supervision of the study was performed by AAA. The manuscript was written and edited by MKN, RB, and AAA.

  *Competing interests.* The authors declare that they have no conflict of interest.

*Acknowledgements.* The authors would like to thank Denis Clement, Jason Dorssers, Katharine McNair, James Stock, Darian Vyriotes, Amanda Pinto, and Phillip Labarge for their contribution in the design of the gondola and thermal camera implementation and analysis. Assistance of Joanne Ryks and Ryan Smith in trial testing of TANAB is appreciated. Useful discussions with John Wilson and Thomas Flesch at the University of Alberta are acknowledged. Useful discussions with Françoise Robe at RWDI are acknowledged. Field support from Michelle Seguin (RWDI), Andrew Bellavie (RWDI), and James Ravenhill at Southern Alberta Institute of Technology (SAIT) is
appreciated. Field assistance from Nick Veriotes is appreciated. The authors are indebted to Steve Nyman, Chris Duiker, Peter Purvis, Manuela Racki, Jeffrey Defoe, Joanne Ryks, Ryan Smith, James Bracken, and Samantha French at the University of Guelph, who helped with the campaign logistics. Special credit is directed toward Amanda Sawlor, Esra Mohamed, Di Cheng, Randy Regan, Margaret Love, Angela Vuk, and Carolyn Dowling-Osborn at the University of Guelph for administrative support. The computational platforms were set up with the assistance of Jeff Madge, Joel Best, and Matthew Kent at the University of Guelph. In-kind technical support for this work was
provided by Rowan Williams Davies and Irwin Inc. (RWDI).

    This work was supported by the Discovery Grant program (401231) from the Natural Sciences and Engineering Research Council (NSERC) of Canada; Government of Ontario through the Ontario Centres of Excellence (OCE) under the Alberta-Ontario Innovation Program (AOIP) (053450); and Emission Reduction Alberta (ERA) (053498). OCE is a member of the Ontario Network of Entrepreneurs (ONE).



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
