# Peer review of "A Tethered And Navigated Air Blimp (TANAB) for observing the microclimate over a complex terrain"

_Geoscientific Instrumentation, Methods and Data Systems, 2019_

## Referee Comment (RC1) · Anonymous Referee #1 · 5 Jun 2019

Review of "A Tethered And Navigated Air Blimp (TANAB) for observing themicroclimate over a complex terrain"

This paper is not suitable for publication in its present form due to two significant short-falls.

1. Lack of suitable literature review outside of the authors' own papers and standard atmospheric boundary-layer (ABL) references. Specifically the wealth of literature on near-surface micro-meteorological observations from tethered systems is missing, be they via kite, tethered balloon or kite-balloon. BL Profile data have been collected in this manner for over a century, and even the more exacting challenge of measuring

[Figure]

ABL turbulence has a history starting in the 1970's. More recently, sonic anemometers (rather than say Gill anenometers) on tethered systems were first made in the 1990's, and in the last 5 years remotely pilot aircraft systems (RPAS) have become the standard platforms for ABL turbulence, and this has been reported in dozens of publications.

2. Lack of instrumental rigour. Both the sonic anemometer data and the remote sensed surface temperatures are not suitably calibrated. A could not find any mention of the means to convert the anemometer wind measurements of [u,v,w] into the reference of the earth: this requires a very significant effort in both analysis of the platform movement (via Inertial Navigation Unit (IMU), not solely from three axis tilt) and calibration of the resulting variances and co-variances. The wind tunnel data are not suitable for this analysis as both units (I assume) were static. Similarly, the over-complex calibration of the FLIR camera's land Surface Temperature (LST) using Plank's Law is inappropriate. These cameras are bolometers, and give an output with a response related to the overall incoming thermal radiation according to Stefan–Boltzmann law, which is far simpler and with fewer coefficients. It is then a matter of calibrating the camera against a known temperature with a known emissivity: using a certificated thermometer in a large lake for instance.

Minor but significant points: when regressing it is best to pivot the data to reduce error.

With such fundamental concerns over the methodology, I am unable to comment on the results.

---

## Referee Comment (RC2) · Anonymous Referee #2 · 15 Jul 2019

TANAB review This paper needs some updates/inclusions/improvements in order to be suitable for publication detailed below. 1) Use of sensors (lack of rigorous or appropriate calibration, and combining of sensor data to allow for appropriate analysis and conclusions to be drawn) a. GPS, there is no mention of ground control, differential or precise point processing (PPP) whichever is the chosen method of processing GPS and thus geolocating the collected data accurately for analysis and interpretation. The author may have carried this out but without a short in calibration of sensors, I cannot comment on the results. b. IMU, there is no detail on the use of the IMU to determine the attitude and motion of the platform this is key to the later calibration and processing of anemometer u,v,w data into the reference of the earth. c. Anemometer, though

wind tunnel calibration had been carried out, it did not detail the means of use of pro-cessed IMU/GPS data to derive the u,v,w reference to the earth. d. FLIR Zenmuse camera is an uncooled VOx microbolometer so reported measurement related to the total incoming IR radiation and requires calibration by means of accurately measuring a ground surface target. Any calibrated ground based temperature sensor measuring one of the ground targets temperature, probably easiest is to measure the lake with a thermometer. 2) Balloons and blimps are not a novel platform and as platforms, there is over a centuries worth of documented use. There is no detail on why the Aero Drum balloon was chosen as opposed to a UAV platform that many of the other researches in this field are choosing to use. I would expect a more in depth review and detail on why the balloon was the best platform, with references to other papers on the use of balloon, kites or UAV as platforms and there use of for observing microclimates. 3) On the choice of field site there needs to be considered and justified arguments why the characteristics of the mine are suitable for study of other microclimate environments.

---

## Author Comment (AC1) · 4 Sep 2019

**A Tethered And Navigated Air Blimp (TANAB) for observing the microclimate over a complex terrain**

**Response to Reviewers**

Nambiar et al.

September 4, 2019

Dear Dr. Mike Rose,

Thank you for the assessment of our manuscript and encouraging submission of a revised manuscript after incorporating reviewer comments. We are pleased to inform you that we have improved the manuscript by addressing all reviewer comments to the best of our ability. Please see below our point-by-point responses. In brief, we have elaborated and reported the wind tunnel calibration of the ultrasonic anemometer in more detail, analyzed the wind measurements in a new *along-wind* reference frame, performed and reported FLIR camera calibration, and enhanced the literature review by adding more text and references relevant to the field. We hope that this new revised version will satisfy the journal requirements and standards. We will be pleased to include any more changes toward the improvement of the manuscript

Regards,

Amir A. Aliabadi

On behalf of all co-authors

Note: The reviewer comments are listed in red font, our responses to reviewer comments are listed in black font, and the changes in the manuscript are listed in blue font.

**1 Reviewer 1**

1. This paper is not suitable for publication in its present form due to two significant shortfalls.

Response: Thank you for your constructive review. The concerns have been addressed to the best of our ability to improve the quality of the manuscript. We hope this revised version will be suitable for publication.

2. Lack of suitable literature review outside of the authors' own papers and standard atmospheric boundary-layer (ABL) references. Specifically the wealth of literature on near-surface micrometeorological observations from tethered systems is missing, be they via kite, tethered balloon, or kite-balloon. BL Profile data have been collected in this manner for over a century, and even the more exacting challenge of measuring ABL turbulence has a history starting in the 1970s.

Response: We appreciate your comments. We have included a detailed literature survey and references in the revised version of the manuscript as the following in Sect. 1.1.

Airborne systems are increasingly being used for atmospheric measurements (Martin et al., 2011; Palomaki et al., 2017) although recently their use is being regulated more restrictively. For instance, rotary or fixed-wing drones are not permitted to fly in complex environments such as busy urban areas and airports. On the other hand, tethered balloon-based atmospheric measurement techniques have been used widely for obtaining the turbulence structure as well as the mean vertical profiles of the ABL in complex environments (Thompson, 1980). One of the main advantages of a tethered balloon system is its ability to profile a significant portion of the planetary boundary layer starting from the surface, which is not possible or economical by ground-based or aircraft-based atmospheric measurement techniques (Egerer et al., 2019). The use of ultrasonic anemometers in tethered balloons have been reported in many studies (Stevens et al., 2013; Canut et al., 2016). In comparison, one of the disadvantage of Pitot tubes is their inability to measure the low wind speeds. So they require a fast flying probe which cannot fly in a complex environment for safety and logistic reasons. Ultrasonic anemometers, on the other hand, are popular because of their continuous measurement characteristics, high accuracy, and their ability to be levitated to measure low velocities (Martikainen et al., 2010; Casten et al., 1995). In addition, studies have shown that tethered balloon-based probes can be used for continuous measurement of the important parameters in the ABL without influencing the flow compared to tower-mounted instruments, where the tower structure can perturb the flow measurement, or presence of complex topography or nearby buildings that can influence the measurement (Haugen et al., 1975; Stevens et al., 2013).

Tethered balloon-borne atmospheric turbulence measurements have a long history of observations over the land (Smith, 1961) and sea (Thompson, 1972) to measure fluxes of heat and moisture at heights up to a few hundred metres. The most notable tethered balloon systems deployed

collected data in campaigns in the late 1960s and 1970s including the Barbados Oceanographic and Meteorological Experiment (BOMEX) (Davidson, 1968; Garstang and La Seur, 1968; Friedman and Callahan, 1970), the Joint Air-Sea Interaction (JASIN) Experiment (Pollard, 1978), and the Global Atmospheric Research Programme (GARP) Atlantic Tropical Experiment (GATE) (Berman, 1976). In BOMEX a tethered balloon system was operated from the deck, which measured temperature, wind, and humidity continuously, at different levels in the range of 0 to 600 m in the ocean area north and east of the Island of Barbados. In JASIN tethered balloons were used to measure the structure of ABL to understand the air-sea interaction in the North Atlantic. In the recent past, tethered balloon systems have been used in Boundary-Layer Late Afternoon and Sunset Turbulence (BLLAST) field campaign that was conducted in southern France (Lothon et al., 2014). Canut et al. (2016) used an ultrasonic anemometer mounted on a tethered balloon system for turbulent flux and variance measurements. Egerer et al. (2019) used the BELUGA (Balloon-bornE moduLarUtility) tethered balloon system for profiling the lower Atmosphere by turbulence and radiation measurements in the Arctic.

3. More recently, ultrasonic anemometers (rather than say Gill anemometers) on tethered systems were first made in the 1990s, and in the last 5 years remotely pilot aircraft systems (RPAS) have become the standard platforms for ABL turbulence, and this has been reported in dozens of publications.

Response: Thank you for your comments. We have incorporated more information about the use of ultrasonic anemometers on airborne measurements and the associated advantages. Please refer to the previous comment.

4. Lack of instrumental rigour. Both the ultrasonic anemometer data and the remote sensed surface temperatures are not suitably calibrated. I could not find any mention of the means to convert the anemometer wind measurements of [u,v,w] into the reference of the earth: this requires a very significant effort in both analysis of the platform movement (via Inertial Navigation Unit (IMU), not solely from three-axes tilt) and calibration of the resulting variances and co-variances. The wind tunnel data are not suitable for this analysis as both units (I assume) were static.

Response: Thank you for pointing out the fact that the ultrasonic anemometer data calibration section were missing important details. Now we have added the gondola motion as well as the calibration experimental set-up in detail in Sect. 2.1 and Sect. 3.1 of the revised manuscript, respectively. We have considered separate calibration equations for both mean and turbulence statistics of the flow. We did not require to convert TriSonica$^{\text{TM}}$ Mini's measurement of flow statistic to a fixed reference frame of the earth. We only required to understand the flow statistics in the *along-wind*, *cross-wind*, and *vertical-wind* directions. As shown below, the gondola turns against the mean wind most of the time and it is level within $\pm 5$ ° in both the pitch and roll angles more than 90% of the time. The gondola only rotates in the yaw direction when the main wind direction changes. This enables us to interpret the flow statistics in the three directions mentioned. Sect. 2.1 of the manuscript now contains the following text.

The gondola is a part of the tethered balloon system controlled by multiple ropes on the ground. The acting forces on the system are 1) the lift force due to the helium-filled balloon, 2) the force of gravity, 3) the tension forces due to the ropes, and 4) the drag forces due to the wind. In the

[Figure]

Figure 1: The wind-rose corresponding to 56 hours of flight, reported in the local coordinate system of the sensor. The numbers on the plot indicate sampling records collected at 10 Hz.

absence of the drag force the remaining forces are acting in the vertical direction and the system ascends up or down, but the presence of the drag force displaces the system in the horizontal direction. At all times these forces are balanced so that the TANAB is in a quasi-stationary position.

The balloon is equipped with a net that helps facing the balloon against the main wind direction at any moment. The net guides the air on one side, and the pressure force stops the balloon from rotating. In case of wind direction changing, the pressure force builds up on the net creating a torque around the centre of rotation that repositions the balloon facing the main wind. Up to three ropes are used to tether the balloon to help with stabilizing the system, especially during high winds associated with convective boundary layers. The ropes, however, impose extra weight on the system so that the vertical range of the system reduces when three ropes are used as opposed to one.

The evidence that the net mechanism succeeds in aligning the balloon against the main wind direction can be found in Fig. 1. The wind-rose shows the wind direction records by TriSonica™ Mini (at TriSonica™ Mini's coordinate) for over 56 hr of flight. According to the wind-rose the gondola mostly faces against the main wind direction because wind direction is recorded from the local north direction most of the time. Note that TriSonica™ Mini's north axis is the sensor local coordinate. If desired, this reading can be converted into the fixed inertial body of reference. For this to occur, the sensor yaw, pitch, and roll angles need to be used. This conversion, however, was not desired in the present analysis.

Figure 2 (left) shows the balloon operation in an unstable atmosphere with high winds when three ropes are used to stabilize it. A sudden drag force on the gondola may drive the system out of its stable position momentarily. Hence it may affect the quality of measurements by creating instabilities. Such phenomenon can be prevented by deploying extra ropes connecting the gondola

[Figure]

[Figure]

Figure 2: TANAB at different atmospheric stability conditions; left: TANAB and three stabilizing ropes deployed at the mining facility; right: TANAB and two stabilizing ropes deployed at the mining facility.

directly to ground operators. The tension in these ropes balances the sudden drag force exerted on the system. This arrangement places the gondola in a quasi-stationary position in the air that indeed helps the stability of measurement in gusty conditions. Figure 2 (right) shows the balloon operation in a stable atmosphere with low winds when only two ropes are used to stabilize it.

A T-connector connects the balloon to the gondola using ropes allowing the gondola to hang freely while minimizing the pitch and roll angles to result in better measurements from a levelled gondola. Figure 3 shows the distributions for pitch and roll angles recorded by TriSonica$^{TM}$ Mini's compass over 56 flight hours. According to the bar charts, the gondola is level within $\pm 5$ ° of pitch and roll angles more than 90% of the time. The gondola only rotates in the yaw direction when the main wind direction changes. The gondola is positioned 2 m below the balloon, so the effects of the balloon motion on the gondola are reduced.

Figure 4 shows the wind tunnel used for ultrasonic anemometer calibration onboard of TANAB, which is an open circuit tunnel designed for turbulent boundary layer research. The reference sensor for calibration was a pitot tube which was used to calibrate an R.M. YOUNG 81000 ultrasonic anemometer and the ultrasonic anemometer onboard of TANAB while all three components of wind were considered from the R.M. YOUNG 81000 ultrasonic anemometer. The azimuth angle, elevation angle, and wind levels were changed, independently, to derive calibration coefficients for both mean and turbulence statistics as measured by the TriSonica$^{TM}$ Mini and calibrated against the R.M. YOUNG 81000. Figures 5 and 6 show the set-up for R.M. YOUNG 81000 and gondola calibration, respectively. Note that the gondola was hung from the tunnel with rope attachments to simulate its motion in the real atmosphere. Section 3.1 of the manuscript now contains the following text.

All the experiments were conducted in the University of Guelph's wind tunnel, which is an open circuit tunnel designed for turbulent boundary layer research. The cross sectional area is 1.2 m $\times$ 1.2 m. The tunnel is 10 m long. The tunnel's air speed is controlled by a gauge that sets the fan speed. The tunnel achieves wind speeds up to 10 m s$^{-1}$. The turbulence intensity is typically

[Figure]

Figure 3: Left: the distribution of pitch angles recorded by TriSonicaTM Mini's compass; right: the distribution of roll angles recorded by TriSonicaTM Mini's compass.

[Figure]

Figure 4: Wind tunnel at the University of Guelph: 1=honeycomb, 2=contraction, 3=test section, 4=diffuser, and 5=fan.

[Figure]

Figure 5: The view of the R.M. YOUNG 81000 (left) and pitot tube (right) at the wind tunnel.

[Figure]

Figure 6: The view of gondola (left) and pitot tube (right) at the wind tunnel.

less than 2 % if no roughness blocks are placed upstream of the flow. The Reynolds number characterizes the turbulence level of the fluid flow and is defined as the ratio of the inertial to viscous forces given by $Re = \frac{\rho \times U \times L}{\mu}$, where $U$ is the flow velocity, $L$ is the characteristic length scale of the system (commonly, the hydraulic diameter of the wind tunnel), and $\mu$ and $\rho$ are the dynamic viscosity and density of the fluid, respectively. In the present study, the wind tunnel's $Re$ number varied between 150,000 and 1,100,000. Considering the size of the wind tunnel, it is capable to generate eddies as large as its physical dimensions.

The performance of the gondola (or the effects of the frame on TriSonica$^{\text{TM}}$ Mini measurements) in reading the mean and turbulence statistics of the flow field is studied with respect to the R.M. YOUNG 81000 anemometer, which is already calibrated, and used for cross comparison to derive the calibration coefficients for the TriSonica$^{\text{TM}}$ Mini using line fits. By adding multiple degrees of freedom, the set-up for this test was designed to further simulate the gondola's movements in real cases. The gondola is attached to the ceiling with tow ropes (featuring the ropes to the balloon) and a single rope to the bottom floor (resembling the ground controller). Now, the gondola faces the main flow (as it does in the real atmosphere), but it has some degrees of freedom to slightly wobble. The azimuth angle, elevation angle, and wind levels were changed, independently, to derive calibration coefficients for both mean and turbulence statistics as measured by the TriSonica$^{\text{TM}}$ Mini and calibrated against the R.M. YOUNG 81000. Both sensors were set up at similar airflow condition while wind speed was varied at few wind levels in the range 2-10 m s$^{-1}$. At each wind speed level, data recording continued for 5 min. Each recording was time averaged to calculate mean and turbulence statistics.

5. Similarly, the over-complex calibration of the FLIR camera's land Surface Temperature (LST) using Plank's Law is inappropriate. These cameras are bolometers, and give an output with a response related to the overall incoming thermal radiation according to Stefan-Boltzmann law, which is far simpler and with fewer coefficients. It is then a matter of calibrating the camera against a known temperature with a known emissivity: using a certificated thermometer in a large lake for instance.

Response: Thank you for your comments regarding the operation and use of the thermal camera onboard the TANAB. The DJI Zenmuse XT was included in the TANAB platform to record Earth surface temperature variation within the mining facility perimeter. Using the default camera constants does not provide absolute surface temperature measurements. Therefore, if one does not calibrate the sensor with known surface temperatures, the relative surface temperatures of the mine, tailings pond, the facility to the East of the mine, and the forest beyond the facility, will be reported. To be able to measure absolute temperatures, we have calibrated the camera constants against known surface temperatures. This method has improved the accuracy of our method. In addition, we added more literature review to address the use of uncooled thermal cameras for numerous similar applications. The following text has been added in the revised manuscript in Sect. 1.1.

From literature, the use of uncooled thermal cameras for quantification of relative surface temperatures have been documented many times for a variety of applications. Improvements in Unmanned Aerial System (UAS) and thermal imaging technology have allowed waterbodies to be mapped with a high spatiotemporal resolution. Specifically, understanding the spatial and temporal variation of thermal plumes from sources, such as influent stormwater or geothermal activity, on waterbodies is of significant importance for many hydrological applications (Baker et al., 2019). Water temperature is also a parameter used to quantify water quality as it is known to impact aquatic organisms (Caldwell et al., 2019). Localized areas of higher surface water temperature were effectively identified by using a DJI Zenmuse XTR uncooled thermal camera without additional radiometric calibration (Caldwell et al., 2019). Mallast and Siebert (2019) utilized a UAS-based FLIR Tau2 thermal camera to spatially map submarine groundwater discharge in the Dead Sea, Israel. The intrusion of cooler groundwater rising to the surface of the Dead Sea was captured by the thermal camera, and the variation of spatial perturbations with respect to time were represented. Similarly, uncooled microbolometer thermal imaging systems have been employed to quantify the spatial distribution of surface temperatures of glaciers (Aubry-Wake et al., 2015). This data can be used to increase the accuracy of energy budget models developed for glaciers (Aubry-Wake et al., 2015). Luo et al. (2018) used a Zenmuse XT uncooled thermal camera to spatially map surface temperature variations of permafrost slopes. Understanding the variation of surface temperature of permafrost slopes is very important for infrastructure management and planning of railways, pipelines, and roads constructed on permafrost (Luo et al., 2018).

Uncooled microbolometer thermal cameras have also been used extensively in the agricultural industry, especially in the emerging precision agriculture sector. Poblete et al. (2018) used a FLIR Tau 2 640 uncooled thermal camera to quantify plant water stress which is a measure of water availability for crops (Alderfasi and Nielsen, 2001). A FLIR Tau 2 640 uncooled microbolometer thermal camera was used by Cao et al. (2018) to image plant leaf temperatures to identify *Sclerotinia sclerotiorum* on the leaves of oilseed rape plants.

UAS-based microbolometer thermal imaging cameras have also been noted to have some niche applications in literature. For example, Murray et al. (2018) successfully conducted a study using the FLIR Systems Vue Pro to identify relative temperature variations between human remains and the surrounding environment. Gallardo-Saavedra et al. (2018) detailed numerous studies where uncooled microbolometer thermal cameras were used to inspect photovoltaic panels and quantify performance of photovoltaic plants. Zhong et al. (2019) completed a study using a FLIR Tau 2 640 to assist in identifying pipeline leakages within a district heating system.

It is well reported in literature that uncooled microbolometer temperature measurements are impacted by variation of sensor and camera/ambient temperatures (FLIR-Systems, 2012; Budzier and Gerlach, 2015; Lin et al., 2018). Considering these three sources of measurement error, it is not acceptable to consider calculated surface temperatures from uncooled microbolometers to be absolute without a calibration. Conversely, cooled thermal cameras are designed to measure temperatures with a high degree of accuracy such that absolute temperatures could be determined (Ribeiro-Gomes et al., 2017). Although cooled thermal cameras are very accurate, they are significantly heavier, more expensive, and require more power to operate (Sheng et al., 2010; Ribeiro-Gomes et al., 2017). As a result, cooled thermal cameras are not commonly used for small UAS (such as drones and blimps) (Sheng et al., 2010).

The following information has been added in Sect. 2.3.1 of the revised manuscript.

Although the camera utilized in this paper used an uncooled microbolometer to record thermal

energy, it was calibrated against known surface temperatures. The use of the Stefan-Boltzmann Law is not applicable because the DJI Zenmuse XT is based on FLIR radiometric thermal imaging technology where the recorded microbolometer value is represented as a signal value comprising energy recorded from the surface, reflected energy from the surface, and atmospheric radiation energy (Zeise et al., 2015). Furthermore, it should be noted that many FLIR radiometric thermal imaging cameras (including the DJI Zenmuse XT) have a 14-bit radiometric resolution capable of recording pixel signal values derived from the camera's A/D converter between 0 and 16383 (FLIR-Systems, 2012; Sagan et al., 2019).[1] The radiometric image pixel signal values can be converted to temperature in Kelvin by performing a radiometric calibration between the recorded pixel signal values and corresponding object surface temperatures (Budzier and Gerlach, 2015). The relationship between the radiometric signal value and the object temperature can be approximated with a Planck curve, as noted by Horny (2003) and similar to Martiny et al. (1996), in Eq. 1 (Budzier and Gerlach, 2015)

$$U_{\text{Obj}} = \frac{R}{\exp\left(\frac{B}{T_{\text{Obj}}}\right) - F} - O,$$
(1)

where $U_{\text{Obj}}$ represents the radiometric pixel signal value, $T_{\text{Obj}}$ represents the surface temperature of the object, $R$ represents the uncooled camera system response, $B$ is a constant derived from Planck's Radiation Law, $F$ accounts for the non-linear nature of the thermal camera system, and $O$ represents an offset (Budzier and Gerlach, 2015). Eq. 1 is rearranged to solve for $T_{\text{Obj}}$ as per Eq. 2 (Budzier and Gerlach, 2015; Tempelhahn et al., 2016)

$$T_{\text{Obj}} = \frac{B}{\ln\left(\frac{R}{U_{\text{Obj}}+O} + F\right)}.$$
(2)

These four values, $R$, $B$, $F$ and $O$ were calculated by the camera manufacturer through the completion of a non-linear regression from the radiometric calibration data (the blackbody surface temperature and the corresponding radiometric pixel signal value) (Budzier and Gerlach, 2015). However, the authors of this study performed a non-linear regression to fit the $R = R_1/R_2$, $B$, $O$, and $F$ constants based on land use type and known surface temperatures in an off-site calibration activity. Note that the complex mining environment did not allow an on-site calibration activity due to access restrictions and safety measures. The raw signal value ($U_{\text{Tot}}$) recorded by the camera is governed by Eq. 3 as described by Usamentiaga et al. (2014)

$$U_{\text{Tot}} = \epsilon\tau U_{\text{Obj}} + \tau(1 - \epsilon)U_{\text{Refl}} + (1 - \tau)U_{\text{Atm}},$$
(3)

where $U_{\text{Obj}}$ is the raw output voltage of a blackbody recorded in a laboratory calibration experiment in the absence of reflection and atmospheric influence in the measured signal. To back calculate $U_{\text{Obj}}$, $U_{\text{Refl}}$ and $U_{\text{Atm}}$ must be determined. $U_{\text{Refl}}$ is the theoretical camera output voltage for a blackbody of temperature $T_{\text{Refl}}$ according to the calibration. $T_{\text{Refl}}$ is the effective temperature of the object surroundings or the reflected ambient temperature. $U_{\text{Atm}}$ is the theoretical raw output voltage of a blackbody based on the assumed atmospheric temperature. $\epsilon$ is the emissivity of the object and $\tau$ is the atmospheric transmissivity. The transmissivity of the atmosphere is generally close to 1.0 (Usamentiaga et al., 2014) under clear weather conditions, so $U_{\text{Atm}}$ does not
* * *
[1] `https://www.dji.com/ca/zenmuse-xt`, last access: 15 February 2019

have to be calculated. From the camera metadata, the assumed reflective temperature ($T_{\text{Refl}}$) was 22 °C. This value was extracted using ExifTool. The $U_{\text{Refl}}$ value was calculated using the same Eq. 1 (Zeise and Wagner, 2016)

$$U_{\text{Refl}} = \frac{R}{\exp\left(\frac{B}{T_{\text{Refl}}}\right) - F} - O, \tag{4}$$

where $R = R_1/R_2$, $B$, $F$ and $O$ are Planck constants of the camera that could be extracted through ExifTool (default constants) or set using fitted constants separately as detailed in Sect. 3.3.

The emissivity of the land surface was determined to be a function of geographic position. The Moderate Resolution Imaging Spectroradiometer (MODIS) instrument MOD11B3 data product was used to derive Land Surface Emissivity (LSE) (Wan et al., 2015). The monthly data product with a resolution of 6 km was used to derive LSE for data collected during the observation campaign in May 2018. The emissivity values were calculated using bands 29, 31, and 32. Since the thermal camera used a Longwave Infrared Radiation (LWIR) detector, radiation within the 7.5 $\mu$m to 13.5 $\mu$m spectral range was included in the camera voltage output.[2] MODIS band 29 records radiation within the 8.4 $\mu$m to 8.7 $\mu$m spectral range, band 31 records radiation within the 10.78 $\mu$m to 11.28 $\mu$m spectral range, and band 32 records radiation within the 11.77 $\mu$m to 12.27 $\mu$m spectral range. These three bands are used in conjunction with the BroadBand Emissivity (BBE) derivation to calculate emissivity as a function of geographical area (Wang et al., 2005). The BBE formula used is

$$BBE = a\epsilon_{29} + b\epsilon_{31} + c\epsilon_{32}, \tag{5}$$

where $a$, $b$, and $c$ are constants that vary based on the land surface material. Wang et al. (2005) determined that the constants $a$, $b$, and $c$ do not vary significantly between soil, vegetation, or anthropogenic materials. However, water, ice, and snow resulted in noticeably different BBE coefficients. Based on Wang et al. (2005), the BBE coefficients for $a$, $b$, and $c$ were chosen to be 0.2122, 0.3859, and 0.4029, respectively. After quantifying $U_{\text{Refl}}$, it is possible to back calculate $U_{\text{Obj}}$ by rearranging Eq. 3 and finally calculate $T_{\text{Obj}}$ via Eq. 2.

The following paragraphs about the calibration are added in Sect. 3.3 of the revised manuscript.

In an attempt to quantify surface temperature measurement inaccuracies of DJI Zenmuse XT, an experiment was conducted to fit the $R$, $B$, $O$ and $F$ parameters as per Eq. 2. Three radiometric images were recorded approximately thirty seconds apart every hour between 0600 Local Daylight Time (LDT) and 2300 LDT over a two day period for four distinct surfaces including water, soil, developed land (urban surfaces), and grass. Each radiometric image captured included a certified thermometer, which measured a corresponding temperature for each land surface. The time delay of approximately thirty seconds between each consecutive image was chosen as Olbrycht and Więcek (2015) noted that uncooled thermal cameras without recent calibration experienced temperature drift as much as 1 °C per minute.

FLIR Tools was used to calculate surface temperatures from the radiometric images recorded by the DJI Zenmuse XT on top of the certified thermometer. For each hourly interval, the average of
* * *
[2]`https://www.dji.com/zenmuse-xt/info/`, last access: 15 February 2019

the surface temperatures from the thermal camera were calculated and were used in the following calculations and figures. The pixel value ($U_{\text{Obj}}$) was calculated using Eq. 1, where constants $R$, $B$, $O$ and $F$ were defined during camera factory calibration and stored in the metadata of each image. As per Table 1 these values are referred to as the default camera constants. The temperatures recorded by the calibrated thermometer were scaled appropriately as the outdoor field test occurred in Guelph, Ontario, Canada which is $334\,\text{m}$ above sea level.

The empirical line method as described by Smith and Milton (1999) was used to relate A/D Counts to the corresponding certified temperatures to calibrate (fit) the $R$, $B$, $O$ and $F$ constants as in Fig. 7. Using the Non-Linear Least-Squares Minimization and Curve-Fitting for Python (LMFIT) library,[3] the constants were fitted and residuals were minimized for each specific surface material imaged during the calibration experiment using Eq. 2. The default and calibrated camera constants are detailed in Table 1. Fig. 7 displays the experimental, default, and calibrated temperatures as a function of camera pixel signal value for water, soil, developed land, and grass, respectively. The non-linear curve fitting library used the certified temperature obtained during the experiment, the corresponding pixel signal value, and Eq. 2 to derive the calibrated camera parameters.

Table 1: Default and calibrated camera constants.

| Camera Parameters | R | B | O | F |
|---|---|---|---|---|
| Default | 366545 | 1428 | −342 | 1 |
| Calibrated Water | 549789 | 1507 | −171 | 1.5 |
| Calibrated Soil | 549800 | 1510 | −171 | 1.5 |
| Calibrated Developed Land | 247614 | 1322 | −513 | 1.5 |
| Calibrated Grass | 314531 | 1391 | −513 | 1.5 |

Fitting of the camera constants for each material resulted in reduced bias and Root Mean Square Error (RMSE) values for water, soil, developed land, and grass, respectively, as compared to the bias and RMSE values considering default camera constants, as shown in Table 2. Gallardo-Saavedra et al. (2018) reported that the manufacturer stated accuracy of the FLIR Vue Pro 640 thermal camera was $\pm 5\,\text{K}$. Kelly et al. (2019) also used the empirical line calibration method for a FLIR Vue Pro 640 thermal camera and determined that the accuracy of the thermal camera was $\pm 5\,\text{K}$, which is in agreement with our findings.

Table 2: Error statistics in temperature measurement associated with default and calibrated camera constants.

| Surface | Water | Soil | Developed Land | Grass |
|---|---|---|---|---|
| Default Bias | 5.18 | 4.81 | 1.83 | 2.07 |
| Default RMSE | 5.83 | 5.34 | 3.91 | 2.34 |
| Calibrated Bias | 0.27 | −0.09 | 0.13 | −0.24 |
| Calibrated RMSE | 2.40 | 1.57 | 3.31 | 1.11 |

The fitted $R$, $B$, $O$ and $F$ constants for water, soil, developed land, and grass were applied to surfaces in the actual mining facility with coordinates corresponding to the closest land use categories. Furthermore, the emissivity of the surface was considered by applying the BBE Eq. 5
* * *
[3]`https://lmfit.github.io/lmfit-py/index.html`, last access: 10 April 2019

(a) Water

(b) Soil

(c) Developed Land

(d) Grass

Figure 7: Certified temperature compared to radiometric image pixel signal value for Water, Soil, Developed Land, and Grass.

derived by Wang et al. (2005). The boundaries for each land use type were determined by visually inspecting the Landsat 8 Operational Land Imager image recorded on May 17, 2018 with a pixel resolution of 30 m. In QGIS, land use type corresponding to geographic coordinates with a spatial resolution of 1 km were applied and the surface temperatures were calculated according to Fig. 8.

The updated land surface temperature maps for each four-hour time period, the corresponding box plots for each four-hour time period, and the comparison of median surface temperatures between the data collected via the TANAB and MODIS on May 24, 2018 are included in Figs. 9, 10, and 11, respectively.

[Figure]

Figure 8: Geographical features of the mining facility as of May 2018.

The updated figures, Fig. 9, 10, and 11 are added in the revised manuscript in Sect. 4.7. In Fig. 11, the resulting maximum relative percentage error in temperature for the corrected pond and mine boundaries and using fitted camera constants was calculated as 3.9%, while the median percentage error was calculated as 0.9%. In comparison to the previous calculation not using correct boundaries and fitted camera constants, these errors were 4.8% and 1.0%, respectively. Clearly, the modified method reduces errors.

6. Minor but significant points: when regressing it is best to pivot the data to reduce error.

Response: We addressed this comment earlier. Please see our response to comment 5 of the same reviewer. For non-linear fitting of the camera constants, the residuals were minimized.

7. With such fundamental concerns over the methodology, I am unable to comment on the results.

Response: Thank you very much for all your comments. We have worked rigorously to incorporate all your suggestions in the revised manuscript. We hope this new version meets the standards of

[Figure]

Figure 9: Median temperature maps for the site at different periods (four-hour intervals) of the day; median temperature calculated for tiles at a resolution of $1\,\text{km} \times 1\,\text{km}$.

[Figure]

Figure 10: Box plot of temperatures for May 2018 tailings pond and mine boundaries for different periods (four-hour intervals) of the day.

[Figure]

(a) TANAB          (b) MODIS          (c) Relative error

Figure 11: Comparison of median temperatures measured by TANAB and MODIS observations in the early afternoon on 24 May 2018; TANAB and MODIS temperatures calculated for tiles at a resolution of $1\,\mathrm{km} \times 1\,\mathrm{km}$.

the journal.

**2 Reviewer 2**

1. This paper needs some updates/inclusions/improvements in order to be suitable for publication detailed below.

Response: Thank you very much for your valuable comments that will help make the content and presentation of the manuscript better. We have performed our best to improve the manuscript based on the comments.

2. Use of sensors (lack of rigorous or appropriate calibration, and combining of sensor data to allow for appropriate analysis and conclusions to be drawn).

Response: We have performed rigorous calibration for the TriSonica$^{TM}$ Mini and the DJI camera. We have addressed this comment. Please see our response to reviewer 1 comment 4 (calibration for the TriSonica$^{TM}$ Mini) and comment 5 (calibration of DJI camera). We have also updated the manuscript rigorously.

3. GPS, there is no mention of ground control, differential or precise point processing (PPP), whichever is the chosen method of processing GPS and thus geolocating the collected data accurately for analysis and interpretation. The author may have carried this out but without a short in calibration of sensors, I cannot comment on the results.

Response: Establishment of physical Ground Control Points (GCP) at the remote mining site was not possible due to the complex nature of the environment. The remote site was actively being excavated during TANAB flights and physical access to desired GCP locations was prohibited. The following three paragraphs has been added to the Sect. 2.3.1 in the revised manuscript.

The N3 flight controller is equipped with two Inertial Measurement Units (IMUs), a Global Navigation Satellite System (GNSS), and Compass system.[4] This flight controller is not capable of Real-Time Kinematic (RTK) positioning.

Post processing of GPS data with Precise Point Positioning (PPP) or RTK methods have been noted to reduce geographical positioning error down to a few centimetres (Satirapod et al., 2003; Remondino et al., 2011; Padró et al., 2019). On the other hand, georeferencing capabilities of GNSS and IMU systems are known to have horizontal geographical positional errors of a few meters (Bláha et al., 2011; Chiabrando et al., 2013; Zhuo et al., 2017). Direct georeferencing without the use of differential corrections provide GPS coordinates of a land point within 2 to 5 m of accuracy (Turner et al., 2014; Whitehead et al., 2014). For the purpose of the TANAB system, direct georeferencing with no corrections were used. Vertical positioning of GNSS/IMU systems have been noted in the literature to be prone to significant errors of up to 50 m especially when compared to other RTK and PPP methods (Eynard et al., 2012; Padró et al., 2019). As a result, vertical positioning of the TANAB gondola was completed by considering the hypsometric equation as per Eq. 6 using the airborne barometric pressure and temperature recorded by the ultrasonic anemometer with respect to ground level barometric pressure at the beginning of each
* * *
[4]http://dl.djicdn.com/downloads/N3/20170825/N3_User_Manual_En_v1.4.pdf, last access: 15 February 2019

TANAB launch.

Even without the use of RTK or PPP, the resulting temperature maps for the surface of the mining facility have a higher spatio-temporal resolution as compared to conventional satellite based sensors. MODIS[5] located on both the Terra and Aqua satellites records two distinct thermal images daily, approximately three hours apart at a $1\,\mathrm{km} \times 1\,\mathrm{km}$ spatial resolution (Crosson et al., 2012; Kumar, 2014; Liu et al., 2017). The Advanced Baseline Imager located on the Geostationary Operational Environmental Satellite (GOES) satellites[6], are capable of capturing thermal images every $5\,\mathrm{min}$ with a spatial resolution of $2\,\mathrm{km} \times 2\,\mathrm{km}$ (Cintineo et al., 2016; Schmit et al., 2017). Furthermore, Landsat satellites are capable of recording Thermal InfRared (TIR) images. The Landsat 7 Enhanced Thematic Mapper Plus[7] can capture TIR images at a spatial resolution of $60\,\mathrm{m} \times 60\,\mathrm{m}$ and the Landsat 8 Thermal Infrared Sensor can record TIR images at a spatial resolution of $100\,\mathrm{m} \times 100\,\mathrm{m}$.[8] Both Landsat satellites have a time resolution of 16 days (Chastain et al., 2019). Although errors are inherently introduced into the directly georeferenced surface temperatures, the advantages of collecting both high spatial and temporal resolution land surface temperature data from the TANAB platform outweigh geographical positioning error from the GNSS/IMU system.

4. IMU, there is no detail on the use of the IMU to determine the attitude and motion of the platform this is key to the later calibration and processing of anemometer u,v,w data into the reference of the earth.

Response: We have addressed this comment earlier as a response to reviewer 1 comment 4 (please see the section about gondola motion). We have added more detail in Sect. 2.1 in the revised manuscript. It was not desired to convert the wind velocity measurements to an earth-fixed reference frame. The system is intended to measure *along-wind*, *cross-wind*, and vertical components of wind velocity vector. Further, it was found that the gondola was mostly levelled within $\pm 5\,°$ 90% of the time and only rotated in the yaw direction. We used IMU roll, pitch, and yaw at a frequency of 200 Hz to quantify the camera line of sight and ultimately performed pixel geographical positioning accordingly. For altitude, we used barometric pressure and temperature measured by the ultrasonic anemometer to calculate the altitude using the standard hypsometric equation (Bolanakis et al., 2015; Stull, 2015).

$$z_2 - z_1 \approx a\overline{T_v}\ln\left(\frac{P_1}{P_2}\right),\tag{6}$$

where $P_1$ and $P_2$ are the pressure measurement at two altitudes $z_1$ and $z_2$. The system measured pressure in mBar although the equation is insensitive to the units of pressure. The unit of altitude is m. $\overline{T_v}$ is the average virtual temperature between altitudes $z_1$ and $z_2$. The constant $a = R_d/g$ is equal to 29.3 m K$^{-1}$ (Stull, 2015). Given the uncertainty of temperature and pressure measurement, the uncertainty of altitude measurement is estimated as 1.2 m.

The uncertainty in altitude calculation is quantified using the theory of error propagation analy-
* * *
[5] https://modis.gsfc.nasa.gov/about/, last access: 20 March 2019

[6] https://www.nasa.gov/content/goes-overview/index.html, last access: 20 February 2019

[7] https://landsat.gsfc.nasa.gov/the-enhanced-thematic-mapper-plus/, last access: 20 February 2019

[8] https://landsat.gsfc.nasa.gov/landsat-8/landsat-8-overview/, last access: 20 February 2019

sis. For the ultrasonic anemometer, the uncertainty of temperature measurement is $2\,K$ and the uncertainty of pressure measurement is $0.01\,kPa$. A sample calculation for the uncertainty of error on the hypsometric equation is detailed below in Eqs. 7, 8 and 9 assuming $P_2 = 100\,kPa$, $P_1 =$ of $101.3\,kPa$, and $\overline{T_v} = 300\,K$ (Ku, 1966). The resulting uncertainty was calculated to be $1.2$ m.

$$\Delta z_2 = \sqrt{\left(\frac{\partial z_2}{\partial \overline{T_v}}\right)^2 \Delta \overline{T_v}^2 + \left(\frac{\partial z_2}{\partial P_2}\right)^2 \Delta P_2^2}, \tag{7}$$

$$\frac{\partial z_2}{\partial \overline{T_v}} = a \ln\left(\frac{P_1}{P_2}\right), \tag{8}$$

$$\frac{\partial z_2}{\partial P_2} = a\overline{T_v}\left(\frac{-1}{P_2}\right). \tag{9}$$

5. Anemometer, though wind tunnel calibration had been carried out, it did not detail the means of use of processed IMU/GPS data to derive the u,v,w reference to the earth.

Response: We have already addressed this comment. Please see our response to reviewer 1 comment 4.

6. FLIR Zenmuse camera is an uncooled VOx microbolometer so reported measurement related to the total incoming IR radiation and requires calibration by means of accurately measuring a ground surface target. Any calibrated ground based temperature sensor measuring one of the ground targets temperature, probably easiest is to measure the lake with a thermometer.

Response: Thank you for your comments. We have addressed this comment in detail. Please see our response to reviewer 1 comment 5.

7. Balloons and blimps are not a novel platform and as platforms, there is over a centuries worth of documented use. There is no detail on why the Aero Drum balloon was chosen as opposed to a Unmanned Aerial Vehicle (UAV) platform that many of the other researchers in this field are choosing to use. I would expect a more in depth review and detail on why the balloon was the best platform, with references to other papers on the use of balloon, kites, or UAV as platforms and their use of for observing microclimates.

Response: Thank you for your comments. We have included a detailed literature review and references in the revised version of the manuscript in Sect. 1.1 (also please see our response to reviewer 1 comment 2.). We reiterate that we are proposing TANAB as a practical, simple, and low cost system for atmospheric turbulence measurements at lower elevation. Aero Drum balloon was chosen because of its low volume round-oval shape that is aerodynamic and requires less helium compared to larger common Zeppelins. We have explained the advantages of tethered balloon over UAVs as response to reviewer 1 comment 3.

8. On the choice of field site there needs to be considered and justified arguments why the characteristics of the mine are suitable for study of other microclimate environments.

Response: Open-pit mining involves altering surface topography and land use at large scales

beyond 10 to 20 km. Unlike urban neighbourhoods at micro scale, open-pit mining has a large footprint beyond the meso scale so meteorological observations require tools that are able to map such scales. e.g. land fills, industrial ponds, farms, dams, reservoirs, and airport. One cannot easily fly a UAV and map such areas. Also there are safety concerns (please see our response to reviewer 1 comment 3) with flying UAVs in such heavily industrialized and complex facilities. This can provide justification why the tethered balloon approach is useful.

**Correspondence:** Amir A. Aliabadi (aliabadi@uoguelph.ca)

**Abstract.** This study presents the first environmental monitoring field campaign of a newly developed Tethered And Navigated Air Blimp (TANAB) system to investigate the microclimate over a complex terrain. The use of a tethered balloon in complex terrains such as mines and tailings ponds is novel and the focus of the present study. The TANAB system was fully developed and launched at a mine facility in northern Canada in May 2018. This study describes the key design features, the sensor payload onboard, calibration, and the observations made by the TANAB system. The system measured meteorological conditions including wind speed in three directions, temperature, relative humidity, and pressure over the first few tens of meters of the atmospheric boundary layer. The system also performed earth surface thermal imaging, or temperature mapping, of the underlying surface. The measurements were made at two primary locations in the facility: i) near a tailings pond and ii) in a mine pit. TANAB measured the dynamics of the atmosphere at different diurnal times (e.g. day versus night) and locations (near tailings pond versus inside the mine). Such dynamics include mean and turbulence statistics pertaining to flow momentum and energy, and they are crucial in the understanding of emission fluxes from the facility in future studies. In addition, TANAB can provide boundary conditions and validation datasets to support mesoscale dispersion modelling or  computational fluid dynamics simulations for various transport models.

**1 Introduction**

The Atmospheric Boundary Layer (ABL) is the lowest portion of the air near the earth surface that responds to surface processes in one hour or less (Stull, 1988; Aliabadi, 2018). The understanding of the atmospheric turbulent processes governing the transfer of heat, moisture, and momentum  in the surface layer is of practical importance for many applications such as weather and climate prediction, pollution dispersion, and urban air quality studies (Zilitinkevich and Baklanov, 2002; Pichugina et al., 2008; Aliabadi et al., 2016b, c). Most of the research in atmospheric turbulence have been focused on relatively smooth terrain and horizontally homogeneous environments mainly due to the limitation in the availability of adequate observation platforms and difficulty in acquiring data from the complex environments. However, the study of the ABL and surface-atmosphere interaction over complex terrain is very important for many applications. Surface heterogeneity can cause horizontal gradients of momentum or temperature, and it can influence or complicate the horizontal and vertical transport mechanisms, for instance driven by slopes or thermals (Mahrt and Vickers,

2005; Medeiros and Fitzjarrald, 2014). In addition, model parameterizations of  turbulent processes established for atmospheric flows over smooth and homogeneous surfaces often fail when applied over inhomogeneous and complex terrains (Roth, 2000).

**1.1 Literature Review**

Two types of ABL observations of the meteorological parameters are key: atmospheric properties and earth surface properties (Mäkiranta et al., 2011; Manoj et al., 2014). Conventional techniques measuring the atmospheric properties such as remote sensing (e.g. satellite, RADARS[1], LIDARS[2], SODARS[3], radiometers) and in situ measurements (meteorological masts, aircraft, or sounding balloons) are widely-used for observing parameters such as wind, humidity, and temperature (Pichugina et al., 2008; Legain et al., 2013; Aliabadi et al., 2016b, 2018a). The main disadvantages of such conventional techniques are the low frequency of turbulence measurements (SODARs and LIDARs), cost (aircraft and satellites), difficulty of navigation (sounding balloons), intermittency of observation (Aircraft, sounding balloons, and non-geostationary satellites), low spatial resolution (geostationary satellites), and limited spatial coverage (meteorological towers) (Fernando and Weil, 2010; Medeiros and Fitzjarrald, 2014). In addition, measuring the surface layer within the ABL poses a serious challenge to aircraft that cannot fly at altitudes lower than 150 m in many jurisdictions for safety reasons (Mayer et al., 2012).

Airborne systems are increasingly being used for atmospheric measurements (Martin et al., 2011; Palomaki et al., 2017) although recently their use is being regulated more restrictively. For instance, rotary or fixed-wing drones are not permitted to fly in complex environments such as busy urban areas and airports. On the other hand, tethered balloon-based atmospheric measurement techniques have been used widely for obtaining the turbulence structure as well as the mean vertical profiles of the ABL in complex environments (Thompson, 1980). One of the main advantages of a tethered balloon system is its ability to profile a significant portion of the planetary boundary layer starting from the surface, which is not possible or economical by ground-based or aircraft-based atmospheric measurement techniques (Egerer et al., 2019). The use of ultrasonic anemometers in tethered balloons have been reported in many studies (Stevens et al., 2013; Canut et al., 2016). In comparison, one of the disadvantage of Pitot tubes is their inability to measure the low wind speeds. So they require a fast flying probe which cannot fly in a complex environment for safety and logistic reasons. Ultrasonic anemometers, on the other hand, are popular because of their continuous measurement characteristics, high accuracy, and their ability to be levitated to measure low velocities (Casten et al., 1995; Martikainen et al., 2010). In addition, studies have shown that tethered balloon-based probes can be used for continuous measurement of the important parameters in the ABL without influencing the flow compared to tower-mounted instruments, where the tower structure can perturb the flow measurement, or presence of complex topography or nearby buildings that can influence the measurement (Haugen et al., 1975; Stevens et al., 2013).

Tethered balloon-borne atmospheric turbulence measurements have a long history of observations over the land (Smith, 1961) and sea (Thompson, 1972) to measure fluxes of heat and moisture at heights up to a few hundred metres. The most notable tethered
* * *
[1]Radio Detection And Ranging
[2]Light Detection And Ranging
[3]Sonic Detection And Ranging

balloon systems deployed collected data in campaigns in the late 1960s and 1970s including the Barbados Oceanographic and Meteorological Experiment (BOMEX) (Davidson, 1968; Garstang and La Seur, 1968; Friedman and Callahan, 1970), the Joint Air-Sea Interaction (JASIN) Experiment (Pollard, 1978), and the Global Atmospheric Research Programme (GARP)

5 Atlantic Tropical Experiment (GATE) (Berman, 1976). In BOMEX a tethered balloon system was operated from the deck, which measured temperature, wind, and humidity continuously, at different levels in the range of 0 to 600 m in the ocean area north and east of the Island of Barbados. In JASIN tethered balloons were used to measure the structure of ABL to understand the air-sea interaction in the North Atlantic. In the recent past, tethered balloon systems have been used in Boundary-Layer Late Afternoon and Sunset Turbulence (BLLAST) field campaign that was conducted in southern France (Lothon et al., 2014).

10 Canut et al. (2016) used an ultrasonic anemometer mounted on a tethered balloon system for turbulent flux and variance measurements. Egerer et al. (2019) used the BELUGA (Balloon-bornE moduLarUtility) tethered balloon system for profiling the lower Atmosphere by turbulence and radiation measurements in the Arctic.

As far as earth surface properties are concerned, Land Surface Temperature (LST) is a crucial meteorological parameter to be measured that influences the energy budget and dynamics of the atmosphere.  LST has historically been de-

15 rived from conventional satellite-based sensors such as the MODerate resolution Imaging Spectroradiometer (MODIS) on the Terra and Aqua satellites, the Advanced Baseline Imager on the Geostationary Operational Environmental Satellites (GOES), the Enhanced Thematic Mapper Plus (ETM+) on Landsat 7 and the Thermal InfRared Sensor (TIRS) on Landsat 8 (Tomlinson et al., 2011; Chastain et al., 2019). These sensors however, only either record images with a high spatial resolution and a low temporal resolution (ETM+ and TIRS) or a high temporal resolution and a low spatial resolution (GOES)

20 (Irons et al., 2012; Cintineo et al., 2016; Schmit et al., 2017). Furthermore, satellite-based sensors are known to have missing and skewed data due to a variety of environmental factors including atmospheric effects, land surface emissivity, and sensor failure (Li et al., 2013; Malamiri et al., 2018).

Recent advancements of Unmanned Aerial Systems (UAS) and miniaturization of thermal imaging technology, have allowed researchers to remotely sense the environment for more reliable and precise LST measurements (Malbéteau

25 et al., 2018). With the inclusion of Inertial Measurement Units (IMU) and Global Navigation Satellite System (GNSS) for UAS, airborne images can be directly georeferenced without Ground Control Points (GCP) (Turner et al., 2014). Quantitative data from images recorded from  UAS can be readily derived from proprietary software including PhotoScan Professional, Pix4Dmapper, and MATLAB (Verykokou and Ioannidis, 2018). Open source direct georeferencing and LST calculation software programs are not commonly used.

30 From literature, the use of uncooled thermal cameras for quantification of relative surface temperatures have been documented many times for a variety of applications. Improvements in UAS and thermal imaging technology have allowed waterbodies to be mapped with a high spatiotemporal resolution. Specifically, understanding the spatial and temporal variation of thermal plumes from sources, such as influent stormwater or geothermal activity, on waterbodies is of significant importance for many hydrological applications (Baker et al., 2019). Water temperature is also a parameter used to quantify water quality as

35 it is known to impact aquatic organisms (Caldwell et al., 2019). Localized areas of higher surface water temperature were effectively identified by using a DJI Zenmuse XTR uncooled thermal camera without additional radiometric calibration

(Caldwell et al., 2019). Mallast and Siebert (2019) utilized a UAS-based FLIR Tau 2 thermal camera to spatially map submarine groundwater discharge in the Dead Sea, Israel. The intrusion of cooler groundwater rising to the surface of the Dead Sea was captured by the thermal camera, and the variation of spatial perturbations with respect to time were represented. Similarly,

5  uncooled microbolometer thermal imaging systems have been employed to quantify the spatial distribution of surface temperatures of glaciers (Aubry-Wake et al., 2015). This data can be used to increase the accuracy of energy budget models developed for glaciers (Aubry-Wake et al., 2015). Luo et al. (2018) used a Zenmuse XT uncooled thermal camera to spatially map surface temperature variations of permafrost slopes. Understanding the variation of surface temperature of permafrost slopes is very important for infrastructure management and planning of railways, pipelines, and roads constructed on permafrost (Luo et al., 2018).

Uncooled microbolometer thermal cameras have also been used extensively in the agricultural industry, especially in the emerging precision agriculture sector. Poblete et al. (2018) used a FLIR Tau 2 640 uncooled thermal camera to quantify plant water stress which is a measure of water availability for crops (Alderfasi and Nielsen, 2001). A FLIR Tau 2 640 uncooled microbolometer thermal camera was used by Cao et al. (2018) to image plant leaf temperatures to identify *Sclerotinia sclerotiorum*

15  on the leaves of oilseed rape plants.

UAS-based microbolometer thermal imaging cameras have also been noted to have some niche applications in literature. For example, Murray et al. (2018) successfully conducted a study using the FLIR Systems Vue Pro to identify relative temperature variations between human remains and the surrounding environment. Gallardo-Saavedra et al. (2018) detailed numerous studies where uncooled microbolometer thermal cameras were used to inspect photovoltaic panels and quantify performance of

20  photovoltaic plants. Zhong et al. (2019) completed a study using a FLIR Tau 2 640 to assist in identifying pipeline leakages within a district heating system.

It is well reported in literature that uncooled microbolometer temperature measurements are impacted by variation of sensor, camera, and ambient temperatures (FLIR-Systems, 2012; Budzier and Gerlach, 2015; Lin et al., 2018). Considering these three sources of measurement error, it is not acceptable to consider calculated surface temperatures from uncooled

25  microbolometers to be absolute without a calibration. Conversely, cooled thermal cameras are designed to measure temperatures with a high degree of accuracy such that absolute temperatures could be determined (Ribeiro-Gomes et al., 2017). Although cooled thermal cameras are very accurate, they are significantly heavier, more expensive, and require more power to operate (Sheng et al., 2010; Ribeiro-Gomes et al., 2017). As a result, cooled thermal cameras are not commonly used for small UAS (such as drones and blimps) (Sheng et al., 2010).

30  ## 1.2  Objectives

There are only a few comprehensive field studies that focus on the ABL over a mine environment while the structure of ABL in an orographically complex terrain such as a mine can be complicated (Rotach and Zardi, 2007; Medeiros and Fitzjarrald, 2014, 2015). In the surface layer, flows are highly influenced by the terrain geometry, while Coriolis effects have still negligible influences (Arroyo et al., 2014). Much less, the authors are only aware of two tethered balloon-based earth surface temperature quantification studies (Vierling et al., 2006; Rahaghi et al., 2019). These gaps in the literature motivated this work.

[revised manuscript text omitted]

(a) TANAB          (b) Schematic of gondola and sensors

**Figure 1.**  Close-up of the Tethered And Navigated Air Blimp (TANAB) system during sampling  and the schematic of the TANAB gondola with all the sensors.

15      The gondola is a part of the tethered balloon system controlled by multiple ropes on the ground. The acting forces on the system are 1) the lift force due to the helium-filled balloon, 2) the force of gravity, 3) the tension forces due to the ropes, and 4) the drag forces due to the wind. In the absence of the drag force the remaining forces are acting in the vertical direction and the system ascends up or down, but the presence of the drag force displaces the system in the horizontal direction. At all times these forces are balanced so that the TANAB is in a quasi-stationary position.
* * *
[4]https://www.rc-zeppelin.com/, last access: 15 February 2019

The balloon is equipped with a net that helps facing the balloon against the main wind direction at any moment. The net guides the air on one side, and the pressure force stops the balloon from rotating. In case of wind direction changing, the pressure force builds up on the net creating a torque around the centre of rotation that repositions the balloon facing the main wind. Up to three ropes are used to tether the balloon to help with stabilizing the system, especially during high winds associated with convective boundary layers. The ropes, however, impose extra weight on the system so that the vertical range of the system reduces when three ropes are used as opposed to one.

The evidence that the net mechanism succeeds in aligning the balloon against the main wind direction can be found in Fig. 2. The wind-rose shows the wind direction records by TriSonica™ Mini (at TriSonica™ Mini's coordinate) for over 56 hr of flight. According to the wind-rose the gondola mostly faces against the main wind direction because wind direction is recorded from the local north direction most of the time. Note that TriSonica™ Mini's north axis is the sensor local coordinate. If desired, this reading can be converted into the fixed inertial body of reference. For this to occur, the sensor yaw, pitch, and roll angles need to be used. This conversion, however, was not desired in the present analysis.

[Figure]

**Figure 2.** The wind-rose corresponding to 56 hours of flight, reported in the local coordinate system of the sensor. The numbers on the plot indicate sampling records collected at 10 Hz.

Figure 3a shows the balloon operation in an unstable atmosphere with high winds when three ropes are used to stabilize it. A sudden drag force on the gondola may drive the system out of its stable position momentarily. Hence it may affect the quality of measurements by creating instabilities. Such phenomenon can be prevented by deploying extra ropes connecting the gondola directly to ground operators. The tension in these ropes balances the sudden drag force exerted on the system. This arrangement places the gondola in a quasi-stationary position in the air that indeed helps the stability of measurement in gusty

conditions. Figure 3b shows the balloon operation in a stable atmosphere with low winds when only two ropes are used to stabilize it.

[Figure]

(a) TANAB and three stabilizing ropes                       (b) TANAB and two stabilizing ropes

**Figure 3.** TANAB at different atmospheric stability conditions tethered with two or three stabilization ropes in the mining facility.

A T-connector connects the balloon to the gondola using ropes allowing the gondola to hang freely while minimizing the pitch and roll angles to result in better measurements from a levelled gondola. Figure 4 shows the distributions for pitch and roll angles recorded by TriSonicaTM Mini's compass over 56 flight hours. According to the bar charts, the gondola is level within $\pm 5\,^\circ$ of pitch and roll angles more than 90 % of the time. The gondola only rotates in the yaw direction when the main wind direction changes. The gondola is positioned 2 m below the balloon, so the effects of the balloon motion on the gondola are reduced.

**2.2 Mini Weather Station**

The TriSonicaTM Mini weather station is an ultrasonic anemometer manufactured by AnemomentTM and is mounted onto the gondola of TANAB[5].[5] This mini weather station is ideal for applications that require a miniature, lightweight, and low velocity anemometer, and are suitable particularly for airborne systems. It has a measurement path length of 35 mm and a weight of 50 gr. The light weight makes it an ideal candidate to use with the TANAB system. It can measure the 3D wind speed, air temperature, relative humidity, and the barometric pressure at a sampling rate up to 10 Hz. The open path provides the least possible distortion of the wind field. Its design with four measurement pathways provides a redundant measurement and the path with the most distortion is removed from the calculations to provide accurate wind measurements. It is also equipped with a compass and a tilt sensor. Because of its low power consumption (only 30 mA at 12 V), it is highly power efficient and can record data for hours.
* * *
[5]https://www.anemoment.com/, last access: 10 January 2019

[Figure]

(a) Pitch angle       (b) Roll angle

**Figure 4.** Distribution of pitch and roll angles recorded by TriSonica$^{TM}$ Mini's compass.

25      A data logger by Applied Technologies Inc.[6] is used as the data synchronization and data collection device for the TriSonica$^{TM}$ Mini weather station. This data logger records the measurements on an SD card onboard that can be retrieved after every flight. In addition, the TriSonica$^{TM}$ Mini can be monitored or programmed using serial communication via the data logger. The TriSonica$^{TM}$ Mini weather station and the data logger are powered using a 12 V Lithium-Polymer (LiPo) battery.

     The anemometer measures wind speed in the range $0 - 30$ m s$^{-1}$ at a resolution of 0.1 m s$^{-1}$. The accuracy of the measurement is $\pm 0.1$ m s$^{-1}$ $(0 - 15$ m s$^{-1})$ or $\pm 2$  % $(15 - 30$ m s$^{-1})$. Wind direction is measured at a resolution of 1 $^\circ$ and an accuracy of $\pm 1$ $^\circ$. Vertical winds are measured appropriately if the approach elevation angle is within $\pm 30$ $^\circ$, a condition that is typically met under calm wind conditions  over smoothly varying topography. Temperature is measured in a range from $-25$ $^\circ$C to $+80$ $^\circ$C with a resolution of 0.1 $^\circ$C and an accuracy of $\pm 2$ $^\circ$C. Pressure is measured in the range $50 - 115$ Pa with

5     an accuracy of $\pm 1$ kPa. The tilt sensor measures the pitch and roll with an accuracy of $\pm 0.5$ $^\circ$. The compass measures the magnetic heading with an accuracy of $\pm 5$ $^\circ$.

**2.3   Thermal Camera and Flight Controller**

The uncooled thermal camera used is the Zenmuse XT, 19-mm lens, manufactured by  DJI.[7] This radiometric camera has a resolution of $640 \times 512$, and it is capable of capturing thermal images at 30 Hz. This camera is powered by DJI

10    and mounted underneath the gondola of the TANAB using a gimbal kit. The camera is operated by the DJI N3 flight controller[8]
* * *
[6]http://www.apptech.com/, last access: 10 March 2019

[7]

[7]https://www.dji.com/ca/zenmuse-xt, last access: 15 February 2019

[8]https://www.dji.com/, last access: 15 February 2019

with associated hardware securely attached to the gondola of the TANAB. This aerial imaging device is designed to be mounted and operated using drone aircraft systems. The camera gimbal system compensates for gondola movement due to wind to keep the camera orientation. The camera system roll angle is generally very close to zero due to the self-stabilization of the device. Nevertheless, the gondola itself is connected to the balloon using ropes and a swivel mechanism to hang freely without being influenced by aerodynamic vibrations of the balloon envelop. The gondola is designed to include a safety feature to protect the camera from impact damage. The DJI Lightbridge 2 (LB2) controller is used with either an iOS or Android device to communicate with the N3 and Zenmuse XT during flight. The LB2 has a maximum communicable range for image transmission of 5 km. Pictures and videos can be recorded with precision while the gondola is in flight as the camera can move independently with respect to the gondola.

During flight, the N3 and DJI Zenmuse XT function in parallel such that the GPS location and altitude of the gondola as well as tilt and heading angles of the camera are recorded by the N3 and are included in the  metadata of each image. All GPS information and associated gondola parameters are recorded and stored within the N3 controller and can be retrieved for later analysis using a mini USB connection. The N3 flight controller is equipped with two IMUs, a GNSS, and Compass system.[9] This flight controller is not capable of Real-Time Kinematic (RTK) positioning.

Post processing of GPS data with Precise Point Positioning (PPP) or RTK methods have been noted to reduce geographical positioning error down to a few centimetres (Satirapod et al., 2003; Remondino et al., 2011; Padró et al., 2019). On the other hand, georeferencing capabilities of GNSS and IMU systems are known to have horizontal geographical positional errors of a few meters (Bláha et al., 2011; Chiabrando et al., 2013; Zhuo et al., 2017). Direct georeferencing without the use of differential corrections provide GPS coordinates of a land point within 2 to 5 m of accuracy (Turner et al., 2014; Whitehead et al., 2014).

For the purpose of the TANAB system, direct georeferencing with no corrections were used. Vertical positioning of GNSS/IMU systems have been noted in the literature to be prone to significant errors of up to 50 m especially when compared to other RTK and PPP methods (Eynard et al., 2012; Padró et al., 2019). As a result, vertical positioning of the TANAB gondola was completed by considering the hypsometric equation as per Eq. 7 using the airborne barometric pressure and temperature recorded by the ultrasonic anemometer with respect to ground level barometric pressure at the beginning of each TANAB launch.

Even without the use of RTK or PPP, the resulting temperature maps for the surface of the mining facility have a higher spatio-temporal resolution as compared to conventional satellite based sensors. The MODIS[10] located on both the Terra and Aqua satellites records two distinct thermal images daily, approximately three hours apart at a 1 km × 1 km spatial resolution (Crosson et al., 2012; Kumar, 2014; Liu et al., 2017). The Advanced Baseline Imager located on the GOES satellites[11], are capable of capturing thermal images every 5 min with a spatial resolution of 2 km × 2 km (Cintineo et al., 2016; Schmit et al., 2017). Furthermore, Landsat satellites are capable of recording Thermal InfRared (TIR) images. The Landsat 7 ETM+[12] can capture TIR images at a spatial resolution of 60 m × 60 m and the Landsat 8 TIRS can record TIR images at a spatial resolution of 100
* * *
[9]http://dl.djicdn.com/downloads/N3/20170825/N3_User_Manual_En_v1.4.pdf, last access: 15 February 2019

[10]https://modis.gsfc.nasa.gov/about/, last access: 20 March 2019

[11]https://www.nasa.gov/content/goes-overview/index.html, last access: 20 February 2019

[12]https://landsat.gsfc.nasa.gov/the-enhanced-thematic-mapper-plus/, last access: 20 February 2019

m $\times$ 100 m.[13] Both Landsat satellites have a time resolution of 16 days (Chastain et al., 2019). Although errors are inherently introduced into the directly georeferenced surface temperatures, the advantages of collecting both high spatial and temporal resolution LST data from the TANAB platform outweigh geographical positioning error from the GNSS/IMU system.

15     The images taken by the thermal camera are saved to an onboard micro SD card. The thermal camera can record images in four different file types including JPEG, R-JPEG, TIFF T-Linear Low, and TIFF T-Linear High. In our study we used R-JPEG. Using the recorded information for each image, the GPS coordinates (latitude and longitude) for individual pixels within each image can be derived. Through using additional software, surface temperature from individual pixels are calculated.

**2.3.1   Image Processing Methodology**

20     The images obtained from field observations were processed utilizing Python (version 3.6), ExifTool (version 10.94), ImageMagick (version 7.07), and mathematical relationships to  LST in Kelvin from pixels for each image. Furthermore, mathematical and trigonometric relationships were employed to directly georeference the image pixels according to the World Geodetic System 1984 (WGS84) datum by deriving decimal degree latitude and longitude values. ExifTool is a software package used to read, write, and edit metadata from images. Within ExifTool, different tags are

25 used depending on the camera manufacturer to extract relevant metadata. Using the FLIR tag in ExifTool, important metadata from each image was derived from the onboard airborne flight controller. ExifTool is executed through the Linux terminal window. ImageMagick is a software used to edit and create images. When extracting the raw data signal value recorded by the thermal camera, ImageMagick is used in conjunction with the ExifTool tag function. ExifTool specifies the data to return from each image and ImageMagick specifies the exact pixel to extract from the image.

30     Through using ExifTool, the following metadata parameters from each image are obtained: latitude of camera gimbal, longitude of camera gimbal, camera gimbal  yaw degree, camera gimbal pitch degree, gondola roll degree, gondola  pitch degree,  date image was recorded, altitude of the gondola (only if the TriSonica[TM] Mini was not operational when the image was captured), the raw signal value recorded by the thermal camera, and the reflected apparent temperature. When airborne, the thermal camera is stabilized, as a result, the camera system is independent of the gondola up to the camera's mechanical range. As a result, images beyond the mechanical extent of the camera were omitted (gondola roll

5 greater than 45 $^\circ$ or less than $-45$ $^\circ$, gondola pitch greater than 45 $^\circ$ and less than $-135$ $^\circ$, and gimbal pitch greater than or equal to 0 $^\circ$, where the recorded pitch angle is located at the  centre of the image, as measured from the horizontal plane). Positive pitch angles primarily include the sky and negative pitch angles primarily include the Earth's surface. Furthermore, camera gimbal pitch angles greater than $-2$ $^\circ$ were removed from the image processing technique as these images were very oblique and would have contributed to LST errors. Camera gimbal pitch angles greater than $-30$ $^\circ$ are known to introduce

10 possible errors into the LST calculation[14]. [14] Since the TANAB was flown up to a maximum of  200 m above
* * *
[13]https://landsat.gsfc.nasa.gov/landsat-8/landsat-8-overview/, last access: 20 February 2019

[14]

[14]https://dl.djicdn.com/downloads/zenmuse_xt/en/sUAS_Radiometry_Technical_Note.pdf, last access: 15 February 2019

ground level, only some images greater than $-30\,^\circ$ were omitted from the image analysis as a compromise between LST spatial distribution and LST accuracy. Additionally, any images with incorrect latitude or longitude values were omitted. For georeferencing simplicity, images recorded with a camera gimbal pitch less than or equal to $-76\,^\circ$ were omitted (including nadir images) to ensure that the pitch angle for the bottom of each image was greater than $-90\,^\circ$. As per the thermal camera specifications, the vertical field of view of the camera is $26\,^\circ$ and the pitch angle for the bottom of an image is equivalent to the camera gimbal pitch angle minus one half of the vertical field of view. This condition was included to avoid negative horizontal distances with respect to the camera/gondola.  Timestamps from each image and the TriSonica$^{\text{TM}}$ Mini data were compared to determine the altitude of the gondola when each image was recorded. The pixel row for images that correspond to sky are calculated such that pixels including sky are omitted.

With georeferencing completed, LST values were derived for every 64th pixel across each row. Pixel rows were selected based on a geometric step function such that the majority of the calculated LST coordinates were located near the top of each image. This  method was chosen as the pixels closer to the top of the image would cover more land surface area.

Although the camera utilized in this paper used an uncooled microbolometer to record thermal energy, it was calibrated against known surface temperatures. The use of the Stefan-Boltzmann Law is not applicable because the DJI Zenmuse XT is based on FLIR radiometric thermal imaging technology where the recorded microbolometer value is represented as a signal value comprising energy recorded from the surface, reflected energy from the surface, and atmospheric radiation energy (Zeise et al., 2015). Furthermore, it should be noted that many FLIR radiometric thermal imaging cameras (including the DJI Zenmuse XT) have a 14-bit radiometric resolution capable of recording pixel signal values derived from the camera's A/D converter between 0 and 16383 (FLIR-Systems, 2012; Sagan et al., 2019).[15] The radiometric image pixel signal values can be converted to temperature in Kelvin by performing a radiometric calibration between the recorded pixel signal values and corresponding object surface temperatures (Budzier and Gerlach, 2015). The relationship between the radiometric signal value and the object temperature can be approximated with a Planck curve, as noted by Horny (2003) and similar to Martiny et al. (1996), in Eq. 1 (Budzier and Gerlach, 2015)

$$U_{\text{Obj}} = \frac{R}{\exp\left(\frac{B}{T_{\text{Obj}}}\right) - F} - O, \tag{1}$$

where $U_{\text{Obj}}$ represents the radiometric pixel signal value, $T_{\text{Obj}}$ represents the surface temperature of the object, $R$ represents the uncooled camera system response, $B$ is a constant derived from Planck's Radiation Law, $F$ accounts for the non-linear nature of the thermal camera system, and $O$ represents an offset (Budzier and Gerlach, 2015). Eq. 1 is rearranged to solve for $T_{\text{Obj}}$ as per Eq. 2 (Budzier and Gerlach, 2015; Tempelhahn et al., 2016)

$$T_{\text{Obj}} = \frac{B}{\ln\left(\frac{R}{U_{\text{Obj}} + O} + F\right)}. \tag{2}$$
* * *
[15]https://www.dji.com/ca/zenmuse-xt last access: 15 February 2019

These four values, $R$, $B$, $F$ and $O$ were calculated by the camera manufacturer through the completion of a non-linear regression from the radiometric calibration data (the blackbody surface temperature and the corresponding radiometric pixel signal value) (Budzier and Gerlach, 2015). However, the  authors of this study performed a non-linear regression to fit the $R = R_1/R_2$, $B$, $O$, and $F$ constants based on land use type and known surface temperatures in an off-site calibration activity. Note that the complex mining environment did not allow an on-site calibration activity due to access restrictions and safety measures. The raw signal  value ($U_{\text{Tot}}$) recorded by the camera is governed by Eq. 3 as described by  Usamentiaga et al. (2014)

$$U_{\text{Tot}} = \epsilon\tau U_{\text{Obj}} + \tau(1 - \epsilon)U_{\text{Refl}} + (1 - \tau)U_{\text{Atm}}, \tag{3}$$

where $$ $U_{\text{Obj}}$ is the raw output voltage of a blackbody recorded  in a laboratory calibration experiment in the absence of reflection and atmospheric influence in the measured signal. To back calculate $U_{\text{Obj}}$, $U_{\text{Refl}}$ and $U_{\text{Atm}}$ must be determined. $U_{\text{Refl}}$ is the theoretical  camera output voltage for a blackbody  of temperature $T_{\text{Refl}}$ according to the calibration. $T_{\text{Refl}}$ is the effective temperature of the object surroundings or the reflected ambient temperature. $U_{\text{Atm}}$ is the theoretical raw output voltage of a blackbody based on the assumed atmospheric temperature. $\epsilon$ is the emissivity of the  object and $\tau$ is the atmospheric transmissivity. The transmissivity of the atmosphere is generally close to 1.0 (Usamentiaga et al., 2014) under clear weather conditions, so $U_{\text{Atm}}$ does not have to be calculated. From the camera metadata, the assumed reflective temperature ($T_{\text{Refl}}$) was 22 °C. This value was extracted using ExifTool. The  $U_{\text{Refl}}$ value was calculated using  the same Eq. 1 (Zeise and Wagner, 2016)

$$U_{\text{Refl}} = \frac{R}{\exp\left(\frac{B}{T_{\text{Refl}}}\right) - F} - O, \tag{4}$$

where $R = R_1/R_2$, $B$, $F$ and $O$ are Planck constants of the camera that could be extracted through ExifTool (default constants) or set using fitted constants separately as detailed in Sect. 3.3.

The  emissivity of the land surface was determined to be a function of geographic position. The  MOD11B3 data product acquired from MODIS was used to derive Land Surface Emissivity (LSE) (Wan et al., 2015). The monthly data product with a resolution of 6 km was used to derive LSE for data collected during the observation campaign in May 2018. The emissivity values were calculated using bands 29, 31, and  32. Since the thermal camera used a Longwave Infrared Radiation (LWIR) detector, radiation within the 7.5 μm to 13.5 μm spectral range was included in the camera voltage output[16]. MODIS band 29 records radiation within the 8.4 μm to 8.7 μm spectral range, band 31 records radiation within the 10.78 μm to 11.28 μm spectral range, and band 32 records radiation within the 11.77 μm to 12.27 μm spectral range. These three bands are used in conjunction with the
* * *
[16]https://www.dji.com/zenmuse-xt/info/, last access: 15 February 2019

Band BroadBand Emissivity (BBE) derivation to calculate emissivity as a function of geographical area (Wang et al., 2005). The BBE formula used is

$$BBE = a\epsilon_{29} + b\epsilon_{31} + c\epsilon_{32}, \tag{5}$$

where $a$, $b$, and $c$ are constants that vary based on the land surface material. Wang et al. (2005) determined that the constants $a$, $b$, and $c$ do not vary significantly between soil, vegetation, or anthropogenic materials. However, water ice , ice, and snow resulted in noticeably different BBE coefficients. Based on Wang et al. (2005), the BBE coefficients for $a$, $b$, and $c$ were chosen to be 0.2122, 0.3859, and 0.4029, respectively.

The output voltage value for the object was then calculated. Finally, the LST value as per the thermal camera was derived using After quantifying $U_{\mathrm{Refl}}$, it is possible to back calculate $U_{\mathrm{Obj}}$ by rearranging Eq. 3 and finally calculate $T_{\mathrm{Obj}}$ via Eq. 2.

**3 Calibration Experiments**

In order to check the validity of the high frequency data from the anemometer onboard of TANAB, the anemometer performance was validated against calibrated sensors prior to the field campaign in a series of wind tunnel and outdoor calibration experiments.

**3.1 Wind Velocity Calibration The mounted anemometer performance was characterized in a highly turbulent flow generated by a wind tunnel at the University of Guelph.**

All the experiments were conducted in the University of Guelph's wind tunnel, which is an open circuit tunnel designed for turbulent boundary layer research. The cross sectional area is 1.2 m $\times$ 1.2 m. The tunnel is 10 m long. The tunnel's air speed is controlled by a gauge that sets the fan speed. The tunnel achieves wind speeds up to 10 m s$^{-1}$. The turbulence intensity is typically less than 2 % if no roughness blocks are placed upstream of the flow. The Reynolds number characterizes the turbulence level of the fluid flow and is defined as the ratio of the inertial to viscous forces given by $Re = \frac{\rho \times U \times L}{\mu}$, where $U$ is the flow velocity, $L$ is the characteristic length scale of the system (commonly, the hydraulic diameter of the wind tunnel), and $\mu$ and $\rho$ are the dynamic viscosity and density of the fluid, respectively. In the present study, the wind tunnel's $Re$ number varied between 150,000 and 1,100,000. Considering the size of the wind tunnel, it is capable of generating eddies as large as its physical dimensions.

The performance of the gondola (or the effects of the frame on TriSonica$^{\mathrm{TM}}$ Mini measurements) in reading the mean and turbulence statistics of the flow field is studied with respect to the R.M. YOUNG 81000 ultrasonic anemometer, which is already calibrated, is and used for cross comparison to derive the calibration coefficients for the anemometer TriSonica$^{\mathrm{TM}}$ Mini using line fits. By adding multiple degrees of freedom, the set-up for this test was designed to further simulate the gondola's movements in the real atmosphere. The gondola is attached to the ceiling with two ropes (featuring the ropes to the balloon) and a single rope to the bottom floor (resembling the ground controller). Now, the gondola faces the main flow (as it does in the real atmosphere), but it has some degrees of freedom to slightly wobble. The azimuth angle, elevation angle, and wind

levels were changed, independently, to derive calibration coefficients for both mean and turbulence statistics as measured by the TriSonica™ Mini and calibrated against the R.M. YOUNG 81000. Both sensors were set up at similar airflow condition while wind speed was varied at  few wind levels in the range  2-10 m s$^{-1}$. At each wind speed level, data recording continued for 5 min. Each recording was time averaged to calculate mean and turbulence statistics.

In this study the velocity along the $X$  $Y$, and $Z$ directions are denoted by $U$, $V$, and $W$. Further, Reynolds decomposition is used to express each velocity component as the sum of the time-averaged and fluctuating components: $U = \overline{U} + u$, $V = \overline{V} + v$, and $W = \overline{W} + w$, where the over-lined quantities are time averages and lower case quantities are instantaneous fluctuations. Furthermore variance and covariances of the fluctuations are represented by $\overline{u^2}$, $\overline{uw}$, etc. The calibration equations obtained are used for correcting the field measurement data from the TANAB.

**3.2 Temperature Calibration**

 Temperatures measured by TriSonica™  Mini are calibrated with respect to the Campbell Scientific HMP60 sensor[17][17] The latter collected minute-averaged temperatures, to which the TriSonica™ Mini temperatures were also averaged and compared. The experiment was carried out under a set of  weather conditions to cover a wide range of temperatures outdoors.

**3.3 Thermal Camera Calibration**

In an attempt to quantify surface temperature measurement inaccuracies of DJI Zenmuse XT, an experiment was conducted to fit the $R$,  $B$, $O$ and $F$ parameters as per Eq. 2. Three radiometric images were recorded approximately thirty seconds apart every hour between 0600 Local Daylight Time (LDT) and 2300 LDT over a two day period for four distinct surfaces including water, soil, developed land (urban surfaces), and grass. Each radiometric image captured included a certified thermometer, which measured a corresponding temperature for each land surface. The time delay of approximately thirty seconds between each consecutive image was chosen as Olbrycht and Więcek (2015) noted that uncooled thermal cameras without recent calibration experienced temperature drift as much as 1 K per minute.

FLIR Tools was used to calculate surface temperatures from the radiometric images recorded by the DJI Zenmuse XT on top of the certified thermometer. For each hourly interval, the average of the surface temperatures from the thermal camera were calculated and were used in the following calculations and figures. The pixel value ($U_{\text{Obj}}$) was calculated using Eq. 1, where constants $R$, $B$, $O$ and $F$ were defined during camera factory calibration and stored in the metadata of each image. As per Table 1 these values are referred to as the default camera constants. The temperatures recorded by the calibrated thermometer were scaled appropriately as the outdoor field test occurred in Guelph, Ontario, Canada which is 334 m above sea level.
* * *
[17] https://www.campbellsci.com/, last access: 10 January 2019

The empirical line method as described by Smith and Milton (1999) was used to relate A/D Counts to the corresponding certified temperatures to calibrate (fit) the $R$, $B$, $O$ and $F$ constants as in Fig. 5. Using the Non-Linear Least-Squares Minimization and Curve-Fitting for Python (LMFIT) library,[18] the constants were fitted and residuals were minimized for each specific surface material imaged during the calibration experiment using Eq. 2. The default and calibrated camera constants are detailed in Table 1. Fig. 5 displays the experimental, default, and calibrated temperatures as a function of camera pixel signal value for water, soil, developed land, and grass, respectively. The non-linear curve fitting library used the certified temperature obtained during the experiment, the corresponding pixel signal value, and Eq. 2 to derive the calibrated camera parameters.

**Table 1.** Default and calibrated camera constants.

| Camera Parameters | R | B | O | F |
|---|---|---|---|---|
| Default | 366545 | 1428 | −342 | 1 |
| Calibrated Water | 549789 | 1507 | −171 | 1.5 |
| Calibrated Soil | 549800 | 1510 | −171 | 1.5 |
| Calibrated Developed Land | 247614 | 1322 | −513 | 1.5 |
| Calibrated Grass | 314531 | 1391 | −513 | 1.5 |

Fitting of the camera constants for each material resulted in reduced bias and Root Mean Square Error (RMSE) values for water, soil, developed land, and grass, respectively, as compared to the bias and RMSE values considering default camera constants, as shown in Table 2. Gallardo-Saavedra et al. (2018) reported that the manufacturer stated accuracy of the FLIR Vue Pro 640 thermal camera was $\pm 5$ K. Kelly et al. (2019) also used the empirical line calibration method for a FLIR Vue Pro 640 thermal camera and determined that the accuracy of the thermal camera was $\pm 5$ K, which is in agreement with our findings.

**Table 2.** Error statistics, bias, and Root Mean Square Error (RMSE), in temperature measurement associated with default and calibrated camera constants.

| Surface | Water | Soil | Developed Land | Grass |
|---|---|---|---|---|
| Default bias | 5.18 | 4.81 | 1.83 | 2.07 |
| Default RMSE | 5.83 | 5.34 | 3.91 | 2.34 |
| Calibrated bias | 0.27 | −0.09 | 0.13 | −0.24 |
| Calibrated RMSE | 2.40 | 1.57 | 3.31 | 1.11 |

The fitted $R$, $B$, $O$ and $F$ constants for water, soil, developed land, and grass were applied to surfaces in the actual mining facility with coordinates corresponding to the closest land use categories. Furthermore, the emissivity of the surface was considered by applying the BBE Eq. 5 derived by Wang et al. (2005). The boundaries for each land use type were determined by visually inspecting the Landsat 8 Operational Land Imager image recorded on May 17, 2018 with a pixel resolution of 30
* * *
[18]https://lmfit.github.io/lmfit-py/index.html, last access: 10 April 2019

(a) Water

(b) Soil

(c) Developed Land

(d) Grass

**Figure 5.** Certified temperature compared to radiometric image pixel signal value for Water, Soil, Developed Land, and Grass.

m. In QGIS, land use type corresponding to geographic coordinates with a spatial resolution of 1 km were applied and the surface temperatures were calculated according to Fig. 6.

**4   Field Experiments and Results**

The TANAB system was launched at a mine facility in northern Canada (above 56 °N) for an environmental monitoring field campaign in May 2018. A schematic of the mine facility can be seen in Fig. 6. The depth of the mine is approximately 100 m. TANAB flew for  56 hr collecting data. The objectives of the measurements were to determine dynamics of the atmosphere at different diurnal times (e.g. day versus night) and locations (near tailings pond versus inside the mine). Such dynamics determine the transport of green house gases (GHGs) and therefore emission fluxes. Measurements of the GHG fluxes were not the objective of this paper and will be addressed elsewhere.

[Figure]

**Figure 6.** A schematic of the mine facility. The black dots represent the outline of the entire facility. The green dots represent the outline of the tailings pond, and the red dots represent the outline of the mine. The blue dots are the balloon launch locations.

Surface level transport mechanisms strongly depend on atmospheric dynamics. Factors such as wind speed, atmospheric diffusion coefficient, and thermal stability greatly influence emission fluxes. As a result, the particular focus of this study is measurement of surface level meteorology in the lowest 200 m  altitude and earth surface temperature. The launch

10   details are summarized in Table 3.

Vertical transport of momentum and heat are predominant processes within the  surface layer of the ABL (Businger et al., 1971) and deriving the vertical fluxes of momentum and heat from wind speed and temperature profile measurements can be achieved using TANAB. Turbulence kinetic energy is one of the key measures of turbulence in the atmosphere as it controls the vertical and horizontal mixing (Lenschow et al., 1980; Svensson et al., 2011; Shin et al., 2013; Canut et al., 2016). It is also used for the parameterization of small-scale turbulent transport processes, such as vertical fluxes, when the smaller scale

5   motions are not modelled directly (Aliabadi et al., 2018b). The equation that represents turbulence kinetic energy is

$$k = \frac{1}{2}(\overline{u^2} + \overline{v^2} + \overline{w^2}), \tag{6}$$

where $\overline{u^2}$, $\overline{v^2}$, and $\overline{w^2}$ are variances of turbulent velocity fluctuations  along-wind ($x$), cross-wind ($y$), and in the vertical ($z$

**Table 3.** Tethered And Navigated Air Blimp (TANAB)  launch details. Times are in Local Daylight Time (LDT).

| Experiment | Location | Start Date | Start time | End time | No. of profiles | Experiment time |
|---|---|---|---|---|---|---|
| 1 | Tailings pond | 2018:05:07 | 21:41:00 | 02:47:00 | 14 | 05:06:00 |
| 2 | Tailings pond | 2018:05:09 | 03:30:00 | 04:00:00 | 02 | 00:30:00 |
| 3 | Tailings pond | 2018:05:10 | 02:30:00 | 08:30:00 | 21 | 06:00:00 |
| 4 | Tailings pond | 2018:05:15 | 04:55:00 | 11:00:00 | 22 | 06:05:00 |
| 5 | Mine | 2018:05:18 | 04:12:00 | 11:12:00 | 20 | 07:00:00 |
| 6 | Mine | 2018:05:19 | 18:52:00 | 23:15:00 | 17 | 04:23:00 |
| 7 | Mine | 2018:05:21 | 11:00:00 | 12:17:00 | 04 | 01:17:00 |
| 8 | Mine | 2018:05:23 | 01:47:00 | 05:30:00 | 10 | 02:43:00 |
| 9 | Mine | 2018:05:24 | 11:19:00 | 14:25:00 | 12 | 03:06:00 |
| 10 | Mine | 2018:05:27 | 14:38:00 | 17:50:00 | 18 | 03:12:00 |
| 11 | Tailings pond | 2018:05:30 | 10:55:00 | 18:57:00 | 24 | 08:02:00 |
| 12 | Tailings pond | 2018:05:31 | 11:07:00 | 14:43:00 | 08 | 03:36:00 |

 directions. We used the hypsometric equation to calculate the altitude of the measurement above surface (Bolanakis et al., 2015; Stull, 2015)

$$z_2 - z_1 \approx a\overline{T_v}\ln\left(\frac{P_1}{P_2}\right),\tag{7}$$

where $P_1$ and $P_2$ are the pressure measurement at two altitudes $z_1$ and $z_2$. The system measured pressure in mBar although the equation is insensitive to the units of pressure. The unit of altitude is m. $\overline{T_v}$ is the average virtual temperature between altitudes $z_1$ and $z_2$. The constant $a = R_d/g$ is equal to 29.3 mK$^{-1}$ (Stull, 2015). Given the uncertainty of temperature and pressure measurement, the uncertainty of altitude measurement is estimated as 1.2 m.

**4.1 Sampling Time**

[revised manuscript text omitted]

In order to quantify the difference in temperatures as a consequence of terrain heterogeneity and other complexities between the mine and tailings pond, we have compared the surface temperatures using box plots in Fig. 18 at different diurnal periods (4-hr intervals) aggregated over the entire observation period as detailed in Table 3. The thermal imaging data is in agreement with the TriSonica$^{TM}$ Mini data as plotted in Fig. 15. For many periods the mine shows higher activity in heat related turbulence statistics such as the turbulent kinematic vertical heat flux during $0800-2000$ LDT and variance of potential temperature during $0000-0400$ LDT. These time intervals correspond to periods when the mine shows a statistically significant higher surface temperature than the pond using the box plots.

The median temperatures measured by TANAB are compared with satellite measurements of MODIS on 24 May 2018. For this comparison, the selected images for TANAB were captured during $1211-1401$ LDT, and for MODIS the selected data source corresponded to 1230 LDT. The horizontal resolution for this comparison was $1\,\mathrm{km}\times1\,\mathrm{km}$. The comparison and the calculated percentage relative error are shown in Fig. 19. The percentage relative error is  3.9 % everywhere within the perimeter of the mining facility and the median relative error within the mining facility is  0.9%.  The LST errors are higher in areas between the mine and the  pond. It appears that TANAB temperature predictions in  this region result from more oblique-angle observations, which are

[Figure]

**Figure 17.** Median temperature maps for the site at different periods (4-hr intervals) of the day; median  temperatures calculated for tiles at a resolution of 1 km × 1 km; times are in Local Daylight Time (LDT).

[Figure]

**Figure 18.**  Box plot of temperatures for tailings pond and mine for different periods (4-hr intervals) of the day; times are in Local Daylight Time (LDT).

prone to a higher error. It should be noted that the land elevation in this region changes drastically from the mine to the pond such that images recorded during the mine launches did not capture LST at the top of this region accurately.

[revised manuscript text omitted]